# WDR62 is required for centriole duplication in spermatogenesis and manchette removal in spermiogenesis

Uda Y. Ho [1]✉, Chun-Wei Allen Feng[1], Yvonne Y. Yeap[1], Amanda L. Bain[2], Zhe Wei [3], Belal Shohayeb [1], Melissa E. Reichelt[1], Hayden Homer [3], Kum Kum Khanna[2], Josephine Bowles[1] & Dominic C. H. Ng [1]✉

WDR62 is a scaffold protein involved in centriole duplication and spindle assembly during mitosis. Mutations in *WDR62* can cause primary microcephaly and premature ovarian insufficiency. We have generated a genetrap mouse model deficient in WDR62 and characterised the developmental effects of WDR62 deficiency during meiosis in the testis. We have found that WDR62 deficiency leads to centriole underduplication in the spermatocytes due to reduced or delayed CEP63 accumulation in the pericentriolar matrix. This resulted in prolonged metaphase that led to apoptosis. Round spermatids that inherited a pair of centrioles progressed through spermiogenesis, however, manchette removal was delayed in WDR62 deficient spermatids due to delayed Katanin p80 accumulation in the manchette, thus producing misshapen spermatid heads with elongated manchettes. In mice, WDR62 deficiency resembles oligoasthenoteratospermia, a common form of subfertility in men that is characterised by low sperm counts, poor motility and abnormal morphology. Therefore, proper WDR62 function is necessary for timely spermatogenesis and spermiogenesis during male reproduction.

[1] School of Biomedical Sciences, Faculty of Medicine, The University of Queensland, Brisbane, QLD, Australia. [2] QIMR Berghofer Medical Research Institute, Brisbane, QLD, Australia. [3] UQ Centre for Clinical Research, Faculty of Medicine, The University of Queensland, Brisbane, QLD, Australia. ✉email: u.ho@uq.edu.au; d.ng1@uq.edu.au

Meiosis is the process where a diploid cell divides twice to produce four haploid gametes[1], whereas a diploid cell divides once in mitosis to generate two diploid cells[2]. The centrosome is the main microtubule organising centre (MTOC) consisting of one pair of centrioles surrounded by pericentriolar matrix (PCM) that anchors the microtubules. The centrioles are replicated alongside DNA during the S-phase and in metaphase they form the spindle poles[2]. However in male meiosis II, the centrioles are predicted to duplicate in the absence of DNA replication[3].

WD40 repeat protein 62 (WDR62) is a 166 kDa scaffold protein that associates with the centrosomes and spindle poles during mitosis. It is the second most commonly mutated gene in autosomal recessive primary microcephaly (MCPH)[4–6]. The N terminus of WDR62 contains multiple WD40 domains that are predicted to form a beta-propeller structure required for microtubule association[7]. In contrast, the C terminus of WDR62 contains serine/threonine phosphorylation motifs used for homodimerisation and heterodimerisation with its paralog, MAPKBP1[8], as well as interaction with kinases such as Aurora Kinase A (AURKA)[9] and c-Jun N-terminal kinase (JNK)[10]. Our recent studies revealed that mitotic AURKA phosphorylation maintains WDR62 localisation at the spindle pole, whereas JNK phosphorylation negatively regulates WDR62 spindle association[7,9].

WDR62 is vital for neuroprogenitor cell maintenance. WDR62 deficiency in mouse models results in abnormal spindle formation, increased mitotic delay, premature differentiation of the radial glial cells into intermediate progenitors and migration delay during brain development[11–14]. Furthermore, reduced AURKA expression and phosphorylation of JNK were also detected in the brains of these mice[11,14]. It has been hypothesised that the loss of WDR62 can exhaust the pool of neuroprogenitor cells required for neurogenesis, leading to thinning of the forebrain cortex and hence microcephaly.

There are 18 MCPH genes identified to date, with the majority associated with centrosome maturation, spindle assembly and cell cycle progression[15]. In addition to neurological defects, these genes are also implicated in meiosis, including meiotic initiation, homologous recombination and spermatocyte cell divisions to form round spermatids. Microcephelin (MCPH1) is required for homologous pairing of chromosomes, meiotic recombination DNA repair and maintaining genome stability in mice[16]. CDK5RAP2 (also known as MCPH3) and ASPM (also known as MCPH5) are required for germ cell maintenance[17,18]. CEP152 (also known as MCPH9) is involved in meiotic spindle formation in the mouse oocyte[19]. CENP-E (also known as MCPH13) is localised to the kinetochores during mitosis and male meiosis[20]. Moreover, centrosomal proteins associated with WDR62, such as CEP63, are similarly required for male meiosis. CEP63-deficient mice are infertile with defects in meiotic recombination, failure in centriole duplication and a complete block in generating mature sperm[21]. Infertility was reported in various Wdr62 genetrap mouse models but this phenotype was not characterised in these initial reports[11–13]. More recent studies revealed that Wdr62 conditional knockout in ovaries and testes resulted in meiotic initiation defect or a delay in germ cell appearance respectively[22], as well as spindle instability and disorientation in the primary spermatocytes[23] and maturing oocytes[24]. This study will further examine the role of WDR62 in sperm development. Our data indicate that WDR62 is involved in centriole duplication during spermatogenesis and in manchette removal during spermiogenesis in mice. Both of these processes are crucial in generating functional spermatozoa for reproduction.

## Results

### Generating a WDR62 depleted mouse model. WDR62 deficiency was produced by genetrap mutagenesis in ES cells supplied by the European Mouse Mutant Cell Repository (ES cell clone: HEPD0788_6_H03). The genetrap cassette was inserted into intron 1 of the Wdr62 gene. The ES cells were injected into JM8A blastocysts to generate chimeric mice and crossed with C57BL/6 mice to generate $Wdr62^{+/tm1a}$ heterozygous mice. $Wdr62^{+/tm1a} \times Wdr62^{+/tm1a}$ breedings produced pups in the approximate ratio of 1:2:1 (23% wild type, 55% heterozygous and 22% homozygous; Supplementary Fig. 1a). Weekly measurement of weight and height for the first 12 weeks after birth revealed normal growth in $Wdr62^{tm1a/tm1a}$ mice compared to control littermates (Supplementary Fig. 1b, c), with a trend towards lighter heart and brain weights in $Wdr62^{tm1a/tm1a}$ (Supplementary Fig. 1d, e).

An analysis of cortical thickness in haematoxylin and eosin (H&E)-stained coronal brain sections showed a significantly thinner cortex in $Wdr62^{tm1a/tm1a}$ P0 brains compared to control littermates (Supplementary Fig. 1f). We further examined the cerebral cortex at P0 by co-staining with SATB2 (layers II–IV) and CTIP2 (layer V–VI) (Supplementary Fig. 1g), demonstrating reduced upper layer II–IV thickness (Supplementary Fig. 1h) and a decreased ratio of SATB2+ (upper layer) to CTIP2+ (lower layer) neurons (Supplementary Fig. 1i). In addition, staining of E14.5 brain sections for PAX6, TBR2 and TBR1 (Supplementary Fig. 1j) showed a decrease in PAX6+ apical progenitors in $Wdr62^{tm1a/tm1a}$ compared to control littermates (Supplementary Fig. 1k). To determine whether the reduced number of apical progenitors was caused by premature cell differentiation, we injected a single pulse of BrdU to pregnant dams 24 h prior to brain isolation at E16.5 and co-stained for Ki67 and BrdU (Supplementary Fig. 1l). Quantification of cells that had exited the cell cycle (BrdU+Ki67−) and were likely differentiating, compared to cells that were actively cycling (BrdU+Ki67+), indicated an approximately 10% increase in cell differentiation in $Wdr62^{tm1a/tm1a}$ cortices compared to control littermates (Supplementary Fig. 1m). We also performed TUNEL staining which showed that there were very few apoptotic cells in the cortex of $Wdr62^{tm1a/tm1a}$ and this was not different to numbers of detectable apoptotic cells in control brains at E14.5 (Supplementary Fig. 1n).

Quantitative real-time PCR (qRT-PCR) analysis of Wdr62 expression in P0 brains showed significantly reduced levels of Wdr62 mRNA transcript in $Wdr62^{tm1a/tm1a}$ (Supplementary Fig. 1o). However, low level of transcripts were detected suggesting our $Wdr62^{tm1a/tm1a}$ mice could potentially be hypomorphs. This was confirmed by WDR62 western analysis, which showed reduced WDR62 protein levels in the $Wdr62^{tm1a/tm1a}$ P0 brain compared to $Wdr62^{+/+}$ and $Wdr62^{+/tm1a}$ littermates (Supplementary Fig. 1p). MAPKBP1 is a paralog and binding partner of WDR62[8]. It is also expressed in the testis and shares similar cellular localisation and interacting partners with WDR62, thus can potentially compensate for WDR62 loss. We used qRT-PCR to examine if Mapkbp1 was upregulated to compensate for WDR62 deficiency. However, similar Mapkbp1 mRNA expression were detected in $Wdr62^{tm1a/tm1a}$ and littermate control P0 brains (Supplementary Fig. 1q), indicating that the expression of Mapkbp1 is not increased to compensate for WDR62 deficiency. Taken together, these data support WDR62 involvement in neuroprogenitor maintenance during brain development and confirmed that our Wdr62 genetrap mouse is consistent with other Wdr62 deficient mouse models[11–13,25].

### WDR62 loss results in smaller testes with less round and elongated spermatids and spermatozoa. To test for fertility, eight sets of $Wdr62^{+/+} \times Wdr62^{+/tm1a}$ controls, four sets of $Wdr62^{+/+}$ (2 females and 2 males) $\times Wdr62^{tm1a/tm1a}$ (2 males

and 2 females) and four sets of $Wdr62^{tm1a/tm1a} \times Wdr62^{tm1a/tm1a}$ matings were set up commencing at 6–8 weeks of age and continuing for 3 months. Litters containing 4–8 pups were obtained from all control matings; however, no pups were obtained from any of $Wdr62^{tm1a/tm1a}$ mating combinations, suggesting that both male and female $Wdr62^{tm1a/tm1a}$ mice were infertile. Therefore, we investigated if ovary development was impaired in our $Wdr62^{tm1a/tm1a}$ female mice by staining for the germ cell marker DEAD-Box helicase 4 (DDX4, also known as mouse VASA homologue, MVH). $Wdr62^{+/+}$ 8-week-old ovaries showed DDX4+ germ cells; however, smaller ovaries without DDX4+ germ cells were observed in $Wdr62^{tm1a/tm1a}$ littermates (Supplementary Fig. 2a, b), indicating an absence of oocytes. This result supports that WDR62 deficient females are unable to reproduce.

Next we examined if testis development was affected in our $Wdr62^{tm1a/tm1a}$ mouse model. Testes were isolated and weighed at postnatal day 28 (P28) and 7 months (adult). We found that $Wdr62^{tm1a/tm1a}$ testes were significantly smaller at both P28 and adult timepoints, when compared to control littermates (Fig. 1a, b and Supplementary Fig. 2c, d). Periodic acid Schiff-haematoxylin (PAS-H) staining of adult testis cross-sections revealed approximately 2.4-fold increase in the percentage of Stage XII seminiferous tubules (Fig. 1d). Further examination of the number of spermatocytes and spermatids at stage III, VII and XII seminiferous tubules showed similar number of primary spermatocytes and spermatocytes in metaphase between $Wdr62^{+/+}$ and $Wdr62^{tm1a/tm1a}$ (Fig. 1e). However, a loss of round spermatids (61% reduction at stage III and 73% reduction at stage VII) and elongated spermatids (71% reduction at stage III, 82% reduction at stage VII and 79% reduction at stage XII) were observed in $Wdr62^{tm1a/tm1a}$ seminiferous tubules (Fig. 1e), suggesting that spermatids were lost possibly during early spermiogenesis. To complement these observations, we conducted qRT-PCR analysis of genes expressed at different stages of spermatogenesis and spermiogenesis. Wdr62 mRNA levels were reduced in the $Wdr62^{tm1a/tm1a}$ testis as expected. Stra8 expression, which indicates meiotic initiation[26], was not different between $Wdr62^{+/+}$ and $Wdr62^{tm1a/tm1a}$ P28 and adult testes. Similarly, we did not find WDR62 deficiency was associated with changes in expression of genes encoding Synaptonemal complex protein 3 (Sycp3), meiosis-specific cohesion protein (Rec8), and recombination protein and DNA meiotic recombinase 1 (Dmc1) markers of early to mid-prophase I[27–29]. There was also no change in the expression of Piwi1, Tdrd5 and Tdrd6, markers of late prophase I of meiosis[30]. Early round spermatids express genes encoding Rfx2, Crem and Sox30 transcription factors that are essential for the progression of spermiogenesis[31–33], we did not observe differences in the mRNA expression of these genes between $Wdr62^{+/+}$ and $Wdr62^{tm1a/tm1a}$ P28 and adult testes, most likely because mRNA transcripts of these genes are produced early in meiosis but are not translated till spermiogenesis[34–36]. In contrast, there was a significantly lower expression of genes encoding Transition protein 1 (Tnp1) and Protamine 2 (Prm2), both involved in compacting the haploid genome in late round spermatids[37], in $Wdr62^{tm1a/tm1a}$ P28 and adult testis (Fig. 1f and Supplementary Fig. 2e) indicating less round spermatids are produced and possible spermiogenesis defect in the $Wdr62^{tm1a/tm1a}$ testis.

**$Wdr62^{tm1a/tm1a}$ testis shows delayed meiotic initiation, prolonged meiosis and apoptosis.** Next, we examined if meiotic initiation was delayed in our $Wdr62^{tm1a/tm1a}$ testis. We used whole mount staining and quantified the number of DDX4+ germ cells co-expressing STRA8, the latter indicating that meiosis has initiated. In $Wdr62^{+/+}$ P5 testis, 80% of the DDX4+ germ

cells also showed STRA8 nuclear localisation. However, approximately 30% of DDX4+ germ cells were also STRA8+ in $Wdr62^{tm1a/tm1a}$ P5 testis indicating reduced or delayed meiotic initiation. In P21 testis sections, 100% of $Wdr62^{+/+}$ seminiferous tubules enclosed DDX4+ germ cells whereas approximately 20% of seminiferous tubules in $Wdr62^{tm1a/tm1a}$ testis contained no DDX4+ germ cells. However, by P28, seminiferous tubules of $Wdr62^{+/+}$ and $Wdr62^{tm1a/tm1a}$ testes showed similar numbers of DDX4+ germ cells (Fig. 2a). Thus our results were consistent with an early defect in meiotic initiation that subsequently recovered by P28 in the WDR62 deficient testis[23]. We also examined the expression of other microcephaly-related genes: Mcph1, Cdk5rap2, and Aspm that were involved in germ cell maintenance[16–18], but we did not detect significant differences in their mRNA expression with WDR62 deficiency (Fig. 2b). This supports the notion that initial defects in meiotic initiation are eventually recovered in $Wdr62^{tm1a/tm1a}$ adult testis.

WDR62 loss-of-function has been shown to cause mitotic delay in mouse embryonic fibroblasts, fibroblasts from patients and during neurogenesis in vivo[11–14]. Therefore we examined whether cell cycle was delayed in prophase I by co-staining for SYCP3, which marks the synaptonemal complex between sister chromatids, and γH2AX, which marks the double-stranded DNA breaks and sex chromosomes in P21, P28 and adult spermatocyte spreads. We quantified the number of cells in the four sub-phases of prophase I: leptotene, zygotene, pachytene and diplotene, as well as in metaphase[38]. Zygotene is characterised by spaghetti-like SYCP3 staining with a cloud of γH2AX staining over the nucleus, as sister chromatids started to pair and double-stranded DNA breaks were presented in preparation for homologous recombination. Pachytene is characterised by short, intense and thick SYCP3 staining, as sister chromatids align and recombine, while γH2AX marks the sex chromosomes. In diplotene, the ends of SYCP3-stained sister chromatids began to unpair, while γH2AX continues to mark the sex chromosomes. SYCP3 and γH2AX localisation in zygotene, pachytene and diplotene was normal and comparable between $Wdr62^{+/+}$ and $Wdr62^{tm1a/tm1a}$ cells (Fig. 2c), suggesting that homologous recombination is normal in $Wdr62^{tm1a/tm1a}$ spermatocytes. Quantitation of the percentage of cells at various stages in P21 testis revealed the presence of metaphase spermatocytes in $Wdr62^{+/+}$, while most of the $Wdr62^{tm1a/tm1a}$ spermatocytes were predominantly in prophase I, between zygotene and diplotene (Fig. 2d). One week later, at P28, both $Wdr62^{+/+}$ and $Wdr62^{tm1a/tm1a}$ spermatocytes showed similar percentages of metaphase spermatocytes (Fig. 2d) most likely indicating that the spermatocytes in the $Wdr62^{tm1a/tm1a}$ testes have caught up to those in the $Wdr62^{+/+}$ testes during the first wave of spermatogenesis. In the adult testis, spermatogenesis is asynchronous, however, no difference in the sub-phases of prophase I was observed between $Wdr62^{+/+}$ and $Wdr62^{tm1a/tm1a}$ (Fig. 2e), indicating that prophase I is unaffected by WDR62 deficiency.

We also examined if cell cycle was delayed by western analysis. Prophase I was marked by γH2AX and RAD51, with the latter involved in double-stranded DNA damage repair by homologous recombination. γH2AX and RAD51 protein expression did not differ between $Wdr62^{+/+}$ and $Wdr62^{tm1a/tm1a}$ in both P28 and adult testis (Fig. 2e) suggesting that prophase I was normal in $Wdr62^{tm1a/tm1a}$ testis. Furthermore, metaphase entry marked by CyclinB1 showed unchanged protein expression in WDR62 deficient testis indicating normal progression to metaphase (Fig. 2e). However, phospho-Histone H3 Ser10 (pHH3), a marker for condensed chromosomes during late G2 and metaphase, was substantially increased in $Wdr62^{tm1a/tm1a}$ P28 and adult testis suggesting possible prolonged metaphase or G2/M arrest (Fig. 2e). We also examined the metaphase to anaphase transition marker,

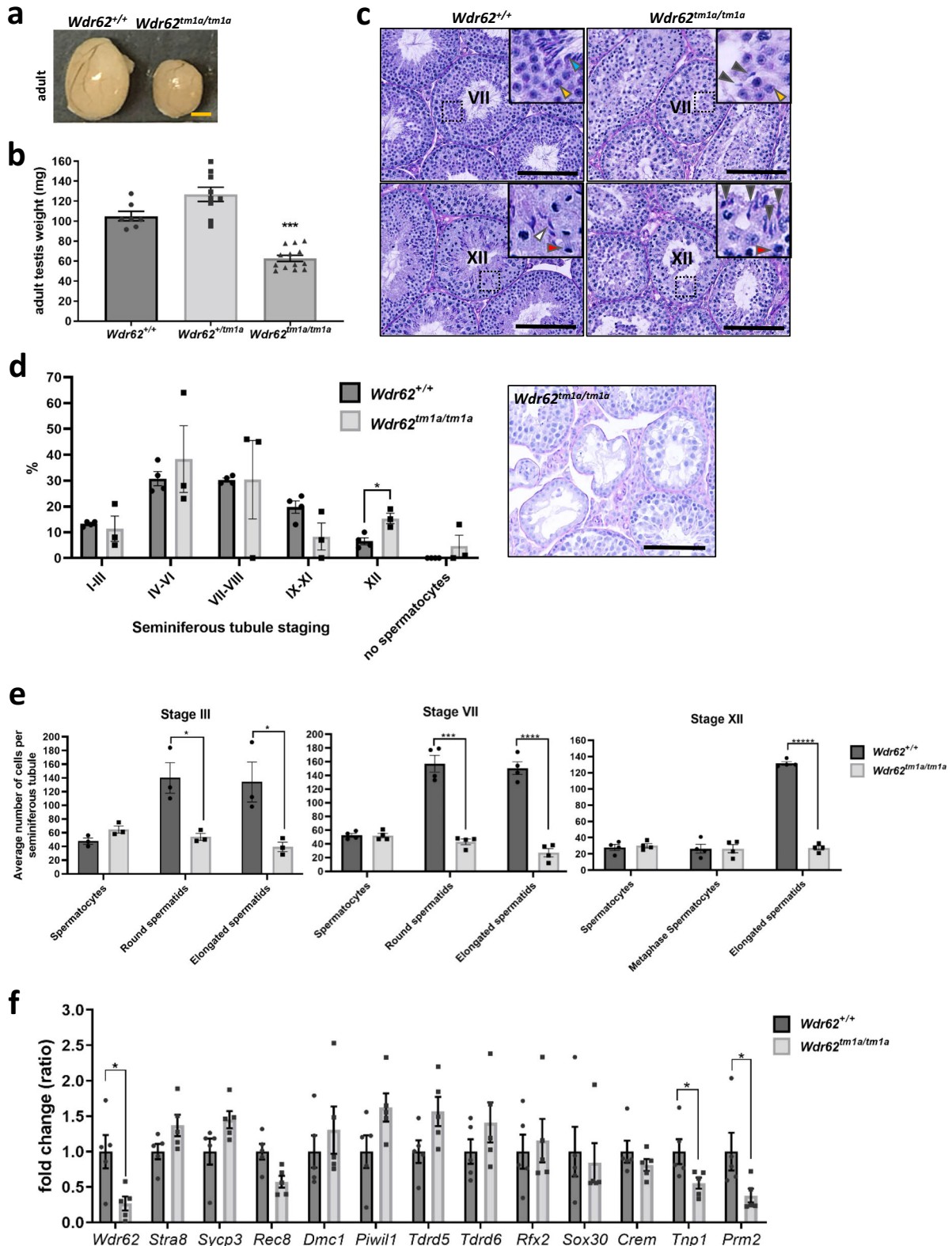

Aurora Kinase B (AURKB), which showed reduced AURKB protein level in *Wdr62*^tm1a/tm1a^ adult testis (Fig. 2e), indicating a decrease in spermatocytes undergoing anaphase. This result correlates well with our observation of reduced numbers of round spermatids in WDR62 deficient testis (Fig. 1e).

To determine if WDR62 deficiency caused a G2/M defect and subsequent apoptosis, we quantified the number of pHH3-positive seminiferous tubules and found a significant increase in the number of pHH3-positive tubules in *Wdr62*^tm1a/tm1a^ P28 and adult testis compared to control littermates (Fig. 3a). To determine if G2/M defect led to apoptosis, we performed TUNEL staining and found an enhanced number of TUNEL-positive tubules in *Wdr62*^tm1a/tm1a^ P28 and adult testis compared to control animals (Fig. 3b). This indicated increased apoptosis in

**Fig. 1 Wdr62$^{tm1a/tm1a}$ testis are smaller with reduced number of round and elongated spermatids. a** Testes from Wdr62$^{+/+}$ and Wdr62$^{tm1a/tm1a}$ littermates collected at 7 months (adult). Scale bar represents 1 mm. **b** Testis weight for Wdr62$^{+/+}$, Wdr62$^{+/tm1a}$ and Wdr62$^{tm1a/tm1a}$ adult testis. $n = 7$ Wdr62$^{+/+}$, 9 Wdr62$^{+/tm1a}$ and 13 Wdr62$^{tm1a/tm1a}$ adult testes. Two-tailed unpaired Student's $t$-test, ****$p = 3.72 \times 10^{-7}$ (see Supplementary Fig. 2c, d for P28 data). **c** Periodic acid Schiff-haematoxylin (PAS-H) staining shows representative stage VII and XII seminiferous tubules in Wdr62$^{+/+}$ and Wdr62$^{tm1a/tm1a}$ adult testis. Insets show 50 μm$^2$ magnification of selected area. Red arrowhead = dividing spermatocyte; yellow arrowhead = round spermatid; white arrowhead = elongated spermatid; blue arrowhead = spermatozoa; grey arrowhead = misshapen spermatid head. Scale bar represents 100 μm. **d** Quantification of PAS-H-stained adult testes sections from b. Inset: PAS-H representative image illustrating seminiferous tubules with an absence of spermatocytes, round spermatids and/or proper elongated spermatids in Wdr62$^{tm1a/tm1a}$ adult testis. $n = 470$ seminiferous tubules from three independent Wdr62$^{+/+}$ adult testes; 373 tubules from three independent Wdr62$^{tm1a/tm1a}$ adult testes. **e** Quantification of the average number of spermatocytes, round spermatids and elongated spermatids per tubule in Wdr62$^{+/+}$ and Wdr62$^{tm1a/tm1a}$ stage III, VII and XII seminiferous tubules. Mean ± SEM. Two-tailed unpaired Student's $t$-test, *$p = 0.020239$ Stage III round spermatids, *$p = 0.034665$ Stage III elongated spermatids, ***$p = 0.000114$ Stage VII round spermatids, ****$p = 0.000031$ Stage VII elongated spermatids, *****$p < 0.000001$ Stage XII elongated spermatids. $N = 15$ Stage III seminiferous tubules from three adult Wdr62$^{+/+}$, 14 Stage III seminiferous tubules from three adult Wdr62$^{tm1a/tm1a}$, 20 Stage VII and XII seminiferous tubules from four adult Wdr62$^{+/+}$, and 20 Stage VII and XII seminiferous tubules from four adult Wdr62$^{tm1a/tm1a}$. **f** qRT-PCR of meiotic initiation (Stra8), spermatogenesis (Sycp3, Rec8, Dmc1, Piwi1, Tdrd5, Tdrd6) and round spermatid (Rfx2, Sox30, Crem, Tnp1, Prm2) markers in Wdr62$^{+/+}$ and Wdr62$^{tm1a/tm1a}$ P28 and adult testes. $N = 3$ Wdr62$^{+/+}$ and 5 Wdr62$^{tm1a/tm1a}$ P28 testes; 5 Wdr62$^{+/+}$ and 5 Wdr62$^{tm1a/tm1a}$ adult testes. Mean ± SEM. Two-tailed unpaired Student's $t$-test, *$p = 0.0210$ (Wdr62), $p = 0.0496$ (Tnp1), $p = 0.0348$ (Prm2) (see Supplementary Fig. 2e for P28 data).

WDR62-deficient testes. We performed SYCP3 and TUNEL co-staining and showed that a majority of TUNEL-positive cells were not prophase I spermatocytes. Co-staining for pHH3 and TUNEL indicated double-positive cells (pHH3+, TUNEL+) in Wdr62$^{tm1a/tm1a}$ but not in Wdr62$^{+/+}$ spermatocytes (Fig. 3c). Thus, WDR62 deficiency leads to meiosis initiation delay, G2/M defect and apoptosis during spermatogenesis.

***Wdr62$^{tm1a/tm1a}$ spermatocytes show spindle misalignment.*** Next, we examined Wdr62 expression during testis development and found that Wdr62 mRNA expression was upregulated at P21 (Supplementary Fig. 3a), a timepoint that coincides with metaphase during the first round of spermatogenesis[39]. On adult spermatocyte spreads, WDR62 was localised to spindles during metaphase and this was absent in Wdr62$^{tm1a/tm1a}$ spermatocytes (Supplementary Fig. 3b). Since conditional Wdr62 knockout in male meiosis has previously reported to show spindle assembly defect and spindle assembly checkpoint (SAC) activation[23], we first performed an analysis of meiotic spindles using α-tubulin staining, which revealed incidences of misaligned meiotic spindles in Wdr62$^{tm1a/tm1a}$ spermatocytes compared to bipolar meiotic spindles with aligned metaphase chromosomes in Wdr62$^{+/+}$ spermatocytes (Supplementary Fig. 3c). This is consistent with a requirement for WDR62 in the proper organisation of meiotic spindles[23].

WDR62 interacts with AURKA and JNK for mitotic spindle regulation[7,9]. Our immunoblot analysis indicated reduced WDR62 protein in P28 and adult testis from Wdr62$^{tm1a/tm1a}$ mice compared to control littermates (Supplementary Figs. 3d and 6b). In contrast to previous studies[11,14,22], AURKA, phosphorylated JNK and total JNK levels were unaltered in Wdr62$^{tm1a/tm1a}$ testes (Supplementary Fig. 3d). We also examined the expression of MAD2, an SAC component that prevents the onset of anaphase until chromosomes were aligned, but no difference was observed between Wdr62$^{tm1a/tm1a}$ and control adult testis (Supplementary Fig. 3e). Therefore, the spindle misalignment observed in WDR62 deficient testis was not due to spindle assembly defect, but rather spindle disorientation.

**WDR62 deficiency in spermatocytes results in centriole underduplication.** Given WDR62 is required for normal centriole duplication[12,40], we determined if there were centriole defects in spermatocytes by performing triple staining of SYCP3, Centrin and Pericentrin in our adult spermatocyte spreads and quantified the number of centrioles in pachytene

and diplotene spermatocytes. Our results revealed approximately 70% of Wdr62$^{+/+}$ pachytene and diplotene spermatocytes contained four centrioles. In contrast, significantly fewer (25%) Wdr62$^{tm1a/tm1a}$ spermatocytes had four centrioles, with 75% containing between zero and three centrioles (Fig. 4a). Thus, our results indicate centriole duplication was impaired in WDR62-deficient spermatocytes.

CEP63 is involved in centriole duplication[21] and is a downstream interacting partner of WDR62[12,40]. We examined whether CEP63 expression and localisation were affected in our Wdr62$^{tm1a/tm1a}$ spermatocytes. Western analysis showed a similar CEP63 protein level between Wdr62$^{+/+}$ and Wdr62$^{tm1a/tm1a}$ P28 and adult testis (Fig. 4b). We also performed SYCP3, Centrin and CEP63 co-staining in adult spermatocyte spreads and quantitated CEP63 intensity, which showed 30% reduced or delayed accumulation of CEP63 at the PCM in Wdr62$^{tm1a/tm1a}$ pachytene/diplotene spermatocytes compared to Wdr62$^{+/+}$ (Fig. 4c, d). Thus, WDR62 is required for effective recruitment of CEP63 to the PCM for centriole duplication, a function conserved between meiosis and mitosis.

CEP63 deficient spermatid heads are misshapen due to potential DNA compaction defects[21]. We also noticed misshapen spermatid heads in our PAS-H-stained Wdr62$^{tm1a/tm1a}$ testis sections at spermiogenesis step 12 (Fig. 1c and Supplementary Fig. 4a). It should be noted that the acrosome (PAS positive staining) seemed to be normal in Wdr62$^{tm1a/tm1a}$ spermatids (Supplementary Fig. 4a)[41], suggesting that WDR62 is not required for acrosome formation. We next sought to determine if centrioles were present in these misshapen spermatid heads. Centrin and Pericentrin co-staining of spermatocyte spreads revealed two centrioles at the neck of the aberrant sperm heads (Fig. 4e [left]). Interestingly, co-staining spermatocyte spreads for Centrin and WDR62 showed WDR62 localisation also at the PCM of Wdr62$^{+/+}$ spermatids (Fig. 4e [right]). Taken together, these data indicated that WDR62 is involved in centriole duplication during meiosis and spermatids that inherited two centrioles (or one centrosome) post-meiosis were able to progress into spermiogenesis.

**WDR62 deficiency leads to oligoasthenoteratospermia.** We examined if viable spermatozoa were generated in the Wdr62$^{tm1a/tm1a}$ mouse model. We performed H&E staining of the epididymis and noticed reduced spermatozoa-filled tubules in Wdr62$^{tm1a/tm1a}$ caput and caudal epididymis (Fig. 5a). Next, we allowed the spermatozoa to swim out from the caudal epididymis and performed sperm counts,

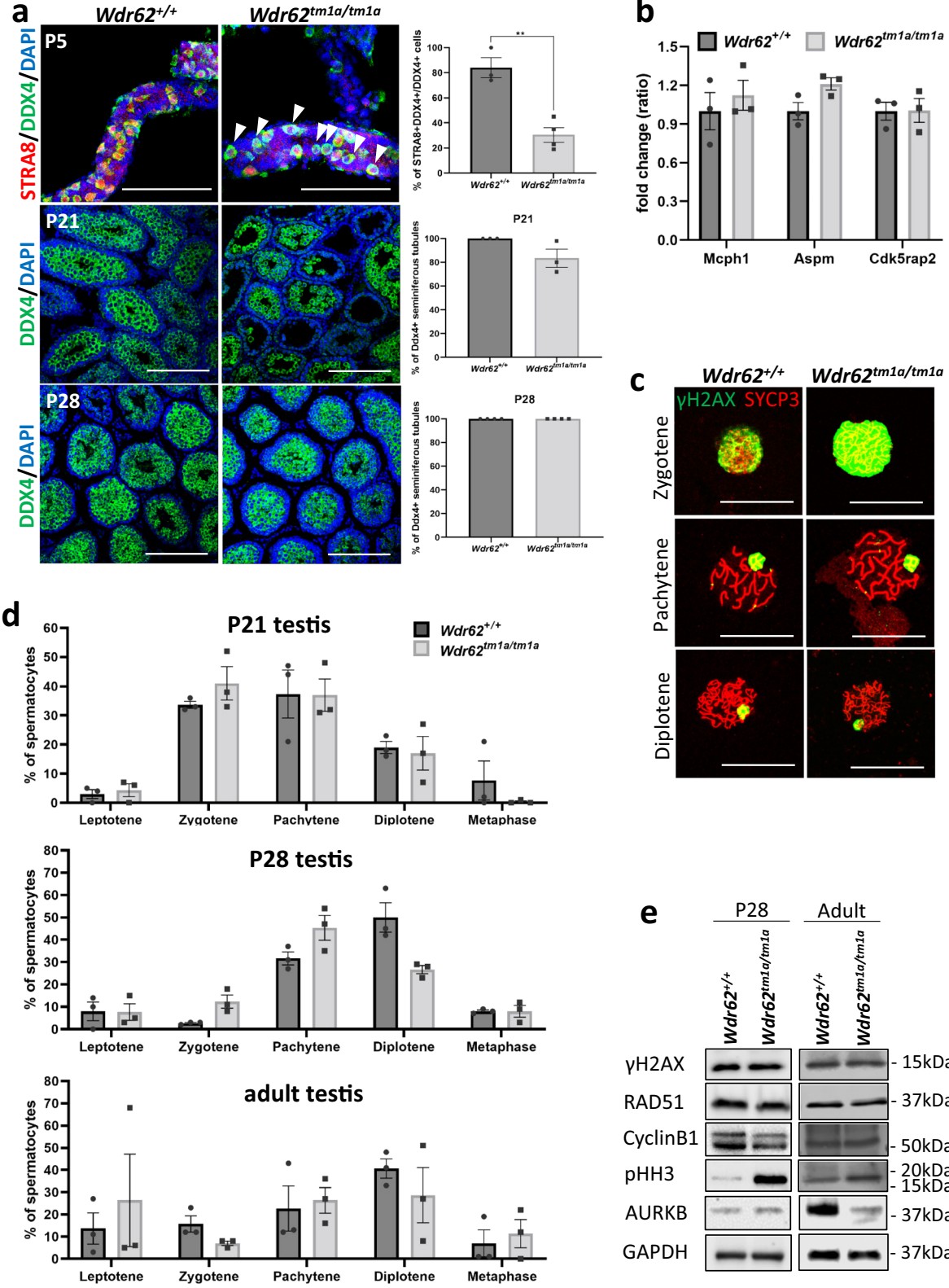

which showed reduced number of spermatozoa in *Wdr62^tm1a/tm1a* indicating oligospermia (Fig. 5b). We also analysed the velocity of these spermatozoa by measuring their progressive movement and noticed that *Wdr62^tm1a/tm1a* sperms swam approximately 50% slower compared to their control littermates (Fig. 5c), indicating partial asthenozoospermia. We further analysed the sperm morphology by performing sperm smear and stain, which showed misshapen sperm

heads in *Wdr62^tm1a/tm1a*, although the axoneme and flagellum appeared normal (Fig. 5d), indicating teratospermia. We have recently reported the involvement of WDR62 in primary cilium formation[25], thus WDR62 deficiency may also affect proper flagellum formation. We performed acetylated α-tubulin immunofluorescence on adult spermatocyte spreads, western analysis of cilium/flagellum elongation markers, CPAP and KIF3β, and transmission electron

**Fig. 2 Delayed meiotic initiation in *Wdr62^tm1a/tm1a* testis. a** DDX4 (green) and STRA8 (red) whole mount immunofluorescence of P5 seminiferous tubules shows meiosis initiated (STRA8+DDX4+) germ cells in *Wdr62^+/+* and *Wdr62^tm1a/tm1a*. However, some DDX4+ cells lack STRA8 nuclear staining in *Wdr62^tm1a/tm1a* tubules (white arrowhead). Quantification of STRA8+DDX4+/total DDX4+ cells indicate reduced meiotic initiation in *Wdr62^tm1a/tm1a* at P5. DDX4 (green) immunofluorescence of P21 and P28 testis sections show DDX4+ germ cells in seminiferous tubules in both in *Wdr62^+/+* and *Wdr62^tm1a/tm1a*. Quantification of DDX4+ tubules/total seminiferous tubules in P21 and P28 testis. $n = 3$ *Wdr62^+/+* and 4 *Wdr62^tm1a/tm1a* P5 testes, 3 *Wdr62^+/+* and 3 *Wdr62^tm1a/tm1a* P21 testes, 3 *Wdr62^+/+* and 3 *Wdr62^tm1a/tm1a* P28 testes. Tubules and sections were counterstained with DAPI (blue). Negative IgG controls are shown in Supplementary Fig. 7a, b. Scale bar represents 200 μm. Two-tailed unpaired Student's $t$-test, $**p = 0.0027$. **b** qRT-PCR shows unaltered Mcph1, Aspm and Cdk5rap2 mRNA expression in *Wdr62^+/+* and *Wdr62^tm1a/tm1a* adult testes. $n = 4$ *Wdr62^+/+* and 4 *Wdr62^tm1a/tm1a* adult testes. **c** γH2AX (green) and SYCP3 (red) co-immunofluorescence on spermatocyte spreads to show different stages of meiosis prophase I. SYCP3 marks the pairing of sister chromatids, whereas γH2AX marks DNA double-strand breaks in leptotene and sex chromosomes in pachytene and diplotene. Scale bar represents 10 μm. **d** The number of leptotene, zygotene, pachytene, diplotene and metaphase spermatocytes from P21, P28 and adult spermatocyte spreads (based on γH2AX and SYCP3 co-staining pattern in Fig. 2c) were counted and graphed as % of total spermatocytes. $n = 332$ *Wdr62^+/+*, 214 *Wdr62^tm1a/tm1a* P21 spermatocytes; 1076 *Wdr62^+/+*, 385 *Wdr62^tm1a/tm1a* P28 spermatocytes; 708 *Wdr62^+/+*, 432 *Wdr62^tm1a/tm1a* adult spermatocytes from three independent pups per genotype per timepoint. Mean ± SEM. **e** Western analysis of P28 and adult testes show unchanged γH2AX, RAD51 and CyclinB1; increased pHH3 and decreased Aurora kinase B (AURKB) protein expression in adult *Wdr62^tm1a/tm1a* testis. GAPDH was used as a loading control. See Supplementary Fig. 6 for western blots and band quantification.

microscopy to examine flagellum formation in WDR62-deficient sperms. These data all showed normal axoneme and flagellum formation in both *Wdr62^+/+* and *Wdr62^tm1a/tm1a* spermatozoa (Supplementary Fig. 4b–d). Hence, WDR62 deficiency in mice leads to low sperm counts, reduced sperm motility and mishappen sperm heads, which are characteristics of oligoasthenoteratospermia, a common cause of subfertility in men.

**WDR62 is required for manchette removal**. Abnormal sperm heads can be caused by defects in DNA compaction/chromatin remodelling and/or manchette assembly and disassembly[42]. The manchette is composed of microtubules and actin filaments. Manchette assembly occurs at steps 7–8 of spermiogenesis, possibly nucleating from the centriolar adjunct associated with the proximal centriole, while manchette disassembly occurs at steps 13–14 of spermiogenesis characterised by zipper-like movement to centriolar adjunct[43]. Since Wdr62 is a microtubule-associated protein[7], we investigated whether Wdr62 was required for manchette removal, using acetylated α-tubulin immunofluorescence on adult spermatocyte spreads. Acetylated α-tubulin was localised to the perinuclear ring and polarised to the caudal side of the spermatid at spermiogenesis steps 7–9 in both *Wdr62^+/+* and *Wdr62^tm1a/tm1a*, suggesting normal manchette formation (Supplementary Fig. 5a). However, from spermiogenesis steps 10–14, elongated manchette was observed in *Wdr62^tm1a/tm1a* spermatids (Fig. 5e and Supplementary Fig. 5a), indicating that Wdr62 was required for efficient manchette disassembly. We confirmed this observation by performing α-tubulin staining in adult testis sections which showed misshapen sperm head and elongated manchette in *Wdr62^tm1a/tm1a* step 12 spermatids compared to *Wdr62^+/+* (Supplementary Fig. 5b). We further investigated the localisation of Pericentrin and CEP63 during manchette formation and deformation. CEP63 was localised similarly to Pericentrin (Supplementary Fig. 5c, d), accumulating at the PCM or the HTCA in step 7–14 *Wdr62^+/+* spermatids. In contrast, CEP63 showed delayed accumulation at the PCM with WDR62 deficiency. CEP63 remained dispersed in the manchette of step 8–10 *Wdr62^tm1a/tm1a* spermatids and did not accumulate in the PCM until steps 10–16 (Supplementary 5d). These data suggest that WDR62 is also required for efficient CEP63 accumulation at the PCM during spermiogenesis, but how CEP63 is involved in this process is still unclear.

The Katanin complex is known for its role in microtubule disassembly. The Katanin p60 subunit (KATNA1) contains ATPase activity for microtubule severing, while the p80 subunit (KATNB1) contains WD domain that interacts with microtubules[44,45]. We therefore examined their expression and

localisation in our WDR62-deficient spermatids during late spermiogenesis. Western blotting showed unaltered KATNA1 and KATNB1 expression in the adult testis (Fig. 5f). Immunofluorescence analysis showed that KATNA1 and KATNB1 were localised to the acrosome/acroplaxome and manchette throughout late spermiogenesis in both *Wdr62^+/+* and *Wdr62^tm1a/tm1a* spermatids (Fig. 6a, b); however, relatively less intense KATNB1-staining in the manchette was seen in step 13–14 *Wdr62^tm1a/tm1a* spermatids compared to *Wdr62^+/+* spermatids (Fig. 6b), suggesting that WDR62 was required for efficient KATNB1 localisation to the manchette during manchette removal. Our studies showed that WDR62–Katanin interactions are required for efficient manchette removal.

## Discussion

In our study, WDR62 deficiency in testes leads to oligoasthenoteratospermia, the most common cause of subfertility in man, characterised by low sperm counts, poor motility and abnormal sperm morphology. At molecular level, the germ cells show impaired centriole duplication, meiotic initiation delay, spindle misalignment, prolonged G2/M phase and enhanced apoptosis, suggesting the role of WDR62 during meiosis reflects similar functions described in mitosis[11–14]. Somatic cells with reduced centriole numbers are known to exhibit delayed spindle assembly, increased duration of prometaphase and overall prolonged mitosis[13,46,47]. Centrosome loss can also trigger p53-dependent apoptosis[46] via the mitosis surveillance pathway USP28-53BP1–p53–p21[48,49]. In mammalian somatic cells, PCM components can accumulate to form MTOC in the absence of centrioles and thus cell division proceeds for a few cycles before arresting[46,47]. However, our data indicate that spermatocytes with reduced centriole number probably underwent prolonged G2/M phase that led to possible p53-dependent apoptosis (without SAC activation) during meiosis, leaving the post-meiotic round spermatids that inherited two centrioles proceeding to spermiogenesis. During spermiogenesis, the proximal centriole associates with the nucleus and forms the centriolar adjunct, while the distal centriole forms the base of the axoneme[43], thereby joining the head and the tail of the sperm. We have shown that post-meiotic *Wdr62^tm1a/tm1a* spermatids and spermatozoa do possess two centrioles (Fig. 4e) with normal axoneme and flagella formation (Fig. 5d Supplementary Fig. 4b–d). Therefore, we have demonstrated that normal centriole number is important in timely spermatogenesis and in subsequent spermiogenesis.

WDR62 recruits CEP63 in a stepwise manner to the centrosome during centriole duplication in in vitro cell lines[40]. We have shown here that reduced accumulation of CEP63 to the

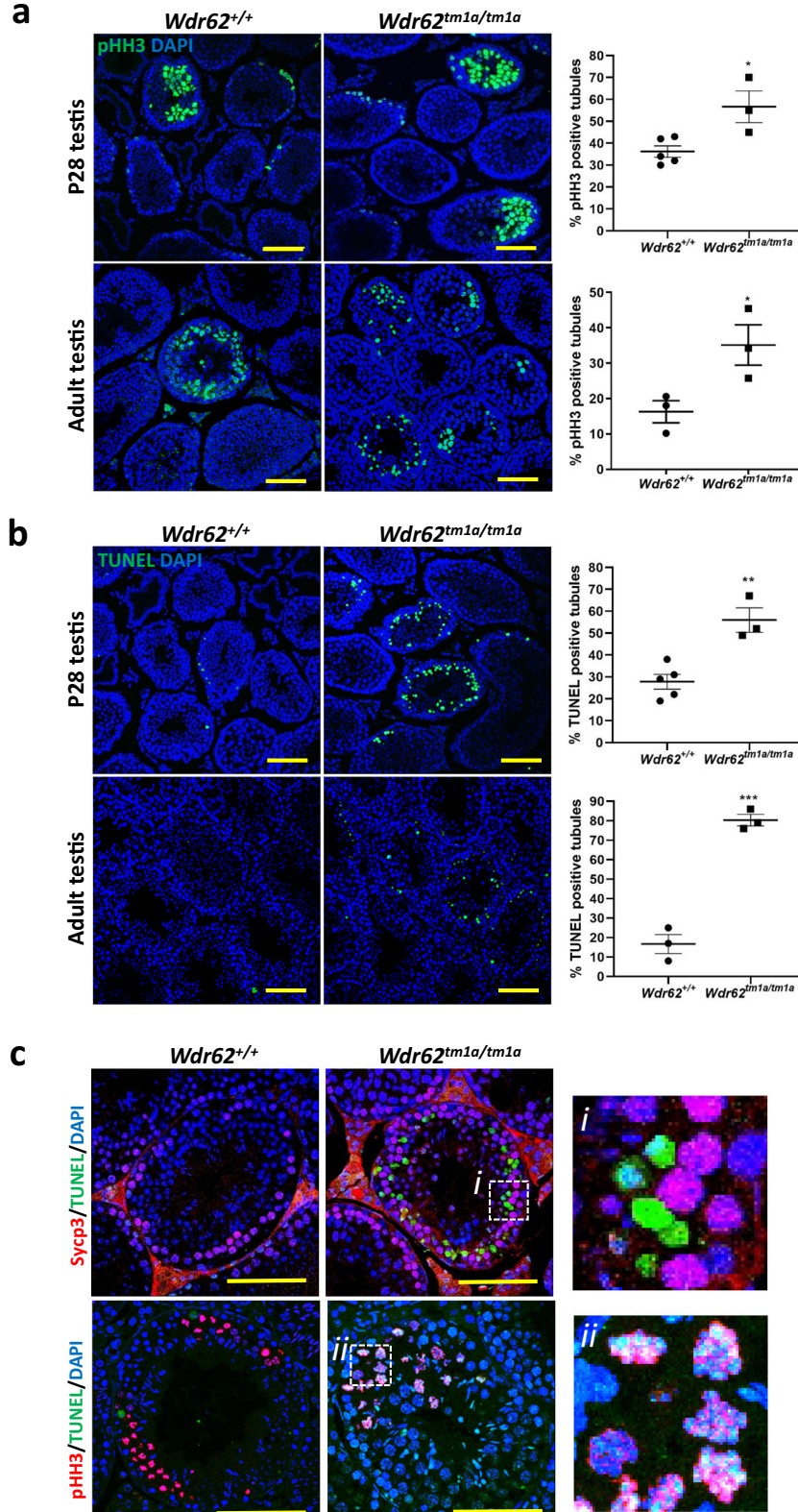

**Fig. 3 Increased metaphase and apoptotic spermatocytes in *Wdr62^tm1a/tm1a* testis. a**. pHH3 (green) immunofluorescence in P28 and adult testis. Negative IgG control is shown in Supplementary Fig. 7c. The percentage of pHH3-positive tubules was quantified. $n = 5$ *Wdr62^+/+* and 3 *Wdr62^tm1a/tm1a* P28 testes; 3 *Wdr62^+/+* and 3 *Wdr62^tm1a/tm1a* adult testes. Two-tailed unpaired Student's *t*-test, $*p = 0.0184$ (P28 testis), $*p = 0.0442$ (adult testis). **b** TUNEL (green) staining in P28 and adult testis. The percentage of TUNEL-positive tubules was quantified. $n = 5$ *Wdr62^+/+* and 3 *Wdr62^tm1a/tm1a* P28 testes; 3 *Wdr62^+/+* and 3 *Wdr62^tm1a/tm1a* adult testes. Two-tailed unpaired Student's *t*-test, $**p = 0.0035$ (P28 testis), $***p = 0.0004$ (adult testis). **c** SYCP3 or pHH3 (red) and TUNEL (green) co-staining in the adult testis. Inset = 50 µm × 50 µm magnified area of dashed box. All sections were counterstained with DAPI (blue). All scale bars represent 100 µm.

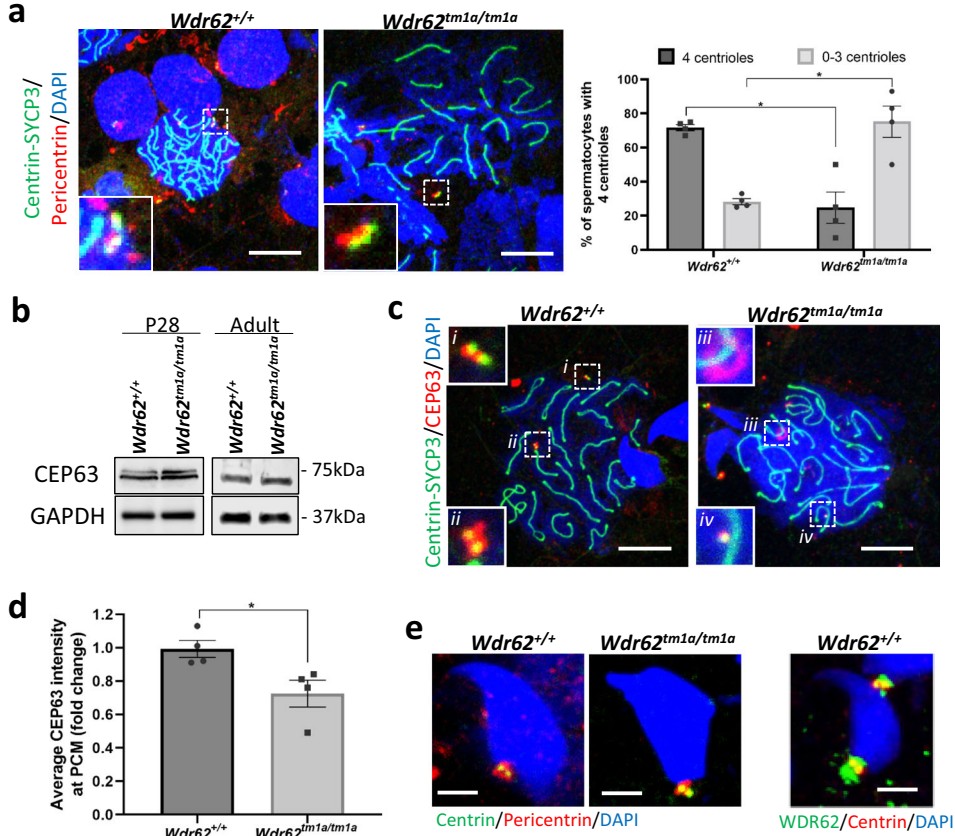

**Fig. 4 *Wdr62^{tm1a/tm1a}* spermatocytes exhibit centriole underduplication during meiosis. a** Centrin-SYCP3 (green) and Pericentrin (red) co-immunofluorescence on spermatocyte spreads from adult testis. Negative IgG control is shown in Supplementary Fig. 7d. The number of centrioles was quantified in pachytene and diplotene spermatocytes and graphed as % of total spermatocytes counted. In *Wdr62^{+/+}* spermatocytes, approximately 70% show four centrioles; however, this percentage is reduced to 25% in *Wdr62^{tm1a/tm1a}* spermatocytes, suggesting centriole underduplication. Scale bar represents 10 μm. *n* = 157 spermatocytes from five independent *Wdr62^{+/+}* and 172 spermatocytes from six independent *Wdr62^{tm1a/tm1a}* testis. Two-tailed Student's *t*-test. *p = 0.0329. **b** CEP63 western blot of P28 and adult *Wdr62^{+/+}* and *Wdr62^{tm1a/tm1a}* testis. GAPDH is used as loading control. See Supplementary Fig. 6 for western blots and band quantification. **c** Centrin-SYCP3 (green) and CEP63 (red) co-immunofluorescence on spermatocyte spreads from adult testis. Scale bar represents 10 μm. Negative IgG controls are shown in Supplementary Fig. 7e. **d** CEP63 intensities from **c** are quantified, showing approximately 20% reduction in CEP63 accumulation at the PCM in *Wdr62^{tm1a/tm1a}* spermatocytes. *n* = 29 *Wdr62^{+/+}* and 26 *Wdr62^{tm1a/tm1a}* pachytene and/or diplotene spermatocytes from three independent testis per genotype. Error bars represent SEM in **a** and **d**. Two-tailed unpaired Student's *t*-test, *p = 0.0305. **e** [Left] Pericentrin (red) and Centrin (green) co-staining of adult *Wdr62^{+/+}* and *Wdr62^{tm1a/tm1a}* spermatocyte spreads. [Right] WDR62 (green) and Centrin (Red) co-immunofluorescence of adult *Wdr62^{+/+}* spermatocyte spreads. A pair of centrioles is localised to the spermatid neck region in both *Wdr62^{+/+}* and *Wdr62^{tm1a/tm1a}* samples. Cells were counterstained with DAPI (blue) in **a**, **c**, **e**. Scale bar represents 5 μm.

centrosomes in *Wdr62^{tm1a/tm1a}* late prophase I spermatocytes have led to centriole underduplication (Fig. 4a–d), indicating that WDR62 is required for efficient CEP63 localisation during meiosis. CEP63 is required for centriole duplication as CEP63 deficiency leads to PCM size reduction, impaired SAS6 recruitment and hence inefficient centriole cartwheel formation during procentriole biogenesis[50]. CEP63 co-localises with CEP152 at the proximal end of the mother centriole[51] which is recruited by CEP57[52]. CEP152 then recruits PLK4 that determines centriole copy number[53], and CPAP/SAS4 to control centriole length[54,55] and attaches the centriole cartwheel to the microtubules[56]. CEP63 is also a mediator of WDR62–ASPM interaction at the proximal end of the mother centriole and the WDR62–CEP63–ASPM complex recruits CPAP/SAS4 for centriole duplication and elongation[12]. It would be interesting to examine if WDR62 also regulates or interacts with CEP57, CEP152 and/or PLK4 recruitment during centriole biogenesis or whether it functions exclusively through interaction with CEP63 in this context.

We were able to examine the role of WDR62 beyond meiosis metaphase I in *Wdr62^{tm1a/tm1a}* mice most likely due to the hypomorphic expression of *Wdr62*. We have shown that WDR62 deficiency leads to misshapen sperm head with elongated manchette. This phenotype is similar to the reported spermatid defects seen in microtubule or IFT related mutants such as Fused[57], Kif3A[58], Ift88[59], and Katanin p80[45]. Indeed, reduced/delayed Katanin p60 and p80 accumulation in the manchette was seen in our WDR62-deficient spermatids at steps 13–14 of spermiogenesis (Fig. 6a, b). WDR62 is predicted to interact with KATNBL1, which is a regulator of the Katanin p60 ATPase-containing subunit responsible for microtubule severing/disassembly[60]. Therefore, it is possible that WDR62 indirectly regulates microtubule severing via the Katanin complexes, since Katanin p60 and p80 subunits are also localised to the spindle poles during mitosis[44] and meiosis as well as the caudal manchette[45]. Our data leave open the possibility that the centriole defect and the manchette defect observed in our WDR62-deficient testis are independent events, as centriole biogenesis and

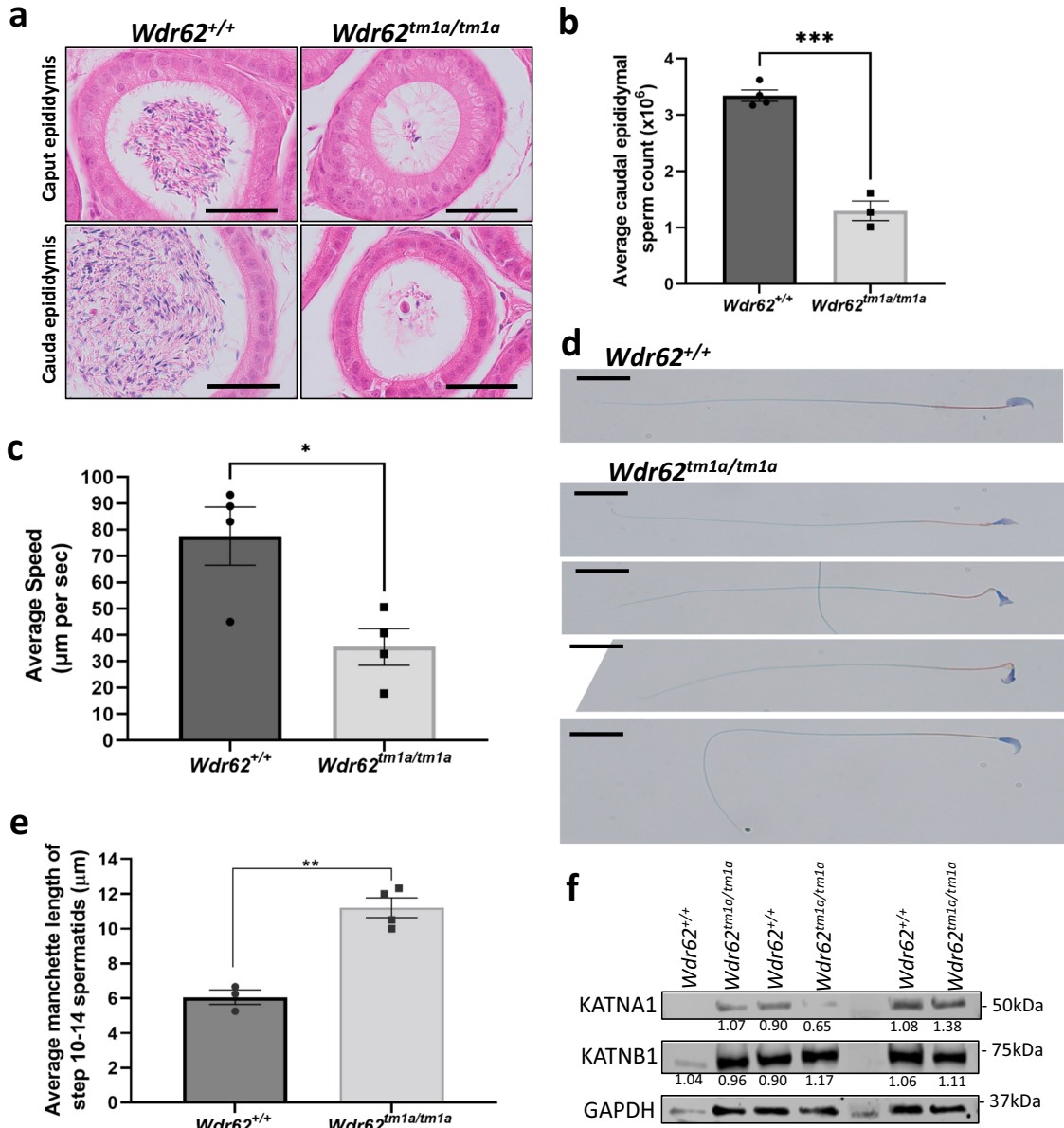

**Fig. 5 Wdr62^tm1a/tm1a spermatozoa show oligoasthenoteratospermia. a** Haematoxylin and eosin staining of adult *Wdr62^+/+* and *Wdr62^tm1a/tm1a* adult epididymis. Scare bar represents 100 μm. **b** Hemocytometer quantification of fixed cauda epididymal spermatozoa shows reduced sperm counts in *Wdr62^tm1a/tm1a*. $n = 4$ *Wdr62^+/+* and 3 *Wdr62^tm1a/tm1a* independent adult mice. Two-tailed unpaired Student's $t$-test, ***$p = 0.0001$. **c** Quantification of sperm motility micrometre swam per second, which showed reduced sperm motility in *Wdr62^tm1a/tm1a* spermatozoa. $N =$ total 190 spermatozoa analysed from four independent *Wdr62^+/+* adult mice; and total 101 spermatozoa analysed from four independent *Wdr62^tm1a/tm1a* adult mice. Two-tailed unpaired Student's $t$-test, *$p = 0.0180$. **d** Sperm stain of adult cauda epididymal sperm smear, which shows normal sperm head (purple), midpiece (orange) and flagellum (green) in *Wdr62^+/+*. However, *Wdr62^tm1a/tm1a* spermatozoa show misshapen sperm heads, with the midpiece and flagellum appear normal. $n = 3$ *Wdr62^+/+* and 3 *Wdr62^tm1a/tm1a* independent mice with representative images of various misshapen sperm heads shown. Scale bar represents 20 μm. **e** Quantification of manchette length (μm) of step 10–14 spermatids (Supplementary Fig. 5a. Acetylated α-tubulin staining of spermatids, which marks the manchette). Mean ± SEM. Two-tailed unpaired Student's $t$-test, **$p = 0.0010$. $n = 11$ *Wdr62^+/+* and 49 *Wdr62^tm1a/tm1a* step 10–14 spermatids from four independent adult mice per genotype. **f** Western analysis shows unaltered Katanin p60 (KATNA1) and Katanin p80 (KATNB1) protein expression in *Wdr62^+/+* and *Wdr62^tm1a/tm1a* adult testis. Normalised band quantification as shown. GAPDH was used as a loading control.

manchette removal occur at different times and stages during sperm development. Although these centrosome-associated proteins may be localised to both the centrioles and manchette, they may have different functions which require further investigations. In addition, we have shown here that WDR62 is required for efficient CEP63 accumulation in the PCM during spermiogenesis (Supplementary Fig. 5d); however, the exact role of CEP63 in spermiogenesis remains unknown. Since CEP63 is required for efficient interaction between WDR62 and ASPM[12] and ASPM directly recruits and binds to KATANIN p60 and p80 subunits during mitosis[61], we propose here that WDR62–CEP63 is required for timely recruitment of ASPM and subsequently KATANIN p60 and p80 to the manchette for microtubule disassembly. Thus the centriolar adjunct could act as a MTOC that anchor the minus end of the microtubule where microtubule severing takes place during manchette disassembly. The relationship between WDR62 and Katanin in microtubule severing will require further studies. Taken together, this study has

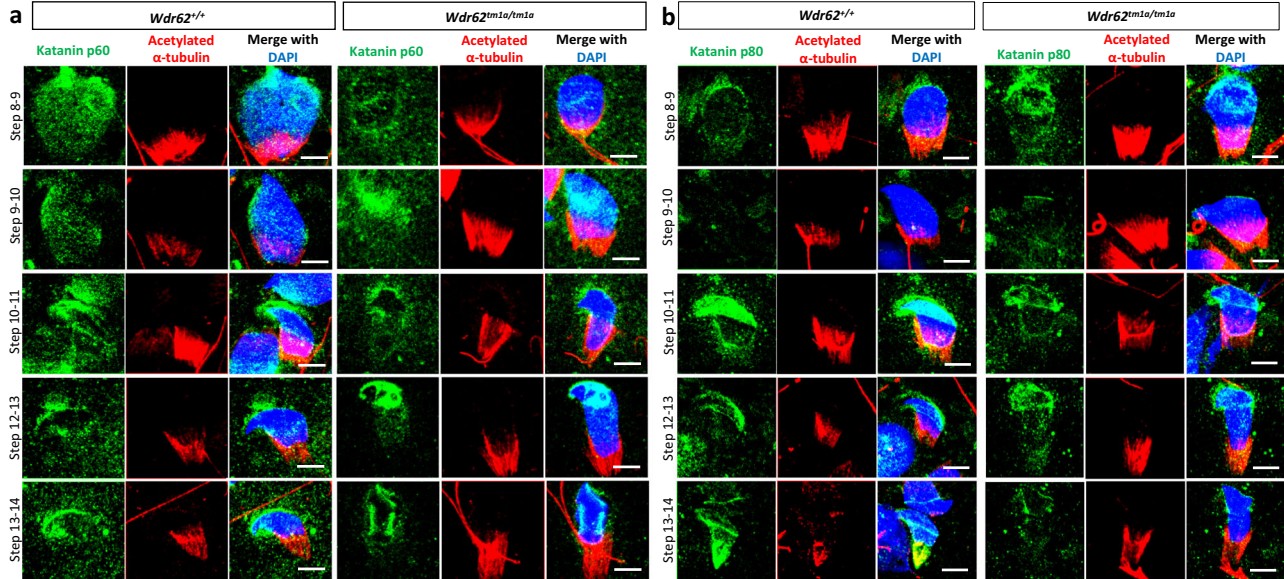

**Fig. 6 Wdr62^tm1a/tm1a spermatids exhibit defective manchette removal. a** Katanin p60 (KATNA1, green) and acetylated α-tubulin (red) co-immunofluorescence on adult spermatocyte spreads. Representative images of co-staining in step 8–14 spermatids are shown. Scale bar represents 5 µm. **b** Katanin p80 (KATNB1, green) and acetylated α-tubulin (red) co-immunofluorescence on adult spermatocyte spreads. Representative images of staining in step 8–14 spermatids are shown. All spermatids are counterstained with DAPI (blue) in (**a**) and (**b**). Scale bars represent 5 µm. Negative IgG controls for (**a**) and (**b**) are shown in Supplementary Fig. 7f.

provided preliminary insight into the relationship between centrosome and manchette removal in meiosis. Future studies should investigate how the centrosome, in particular the relationship between WDR62 and CEP63 and/or Katanin and other centrosomal/cilia-associated proteins, are involved in manchette formation and deformation during spermiogenesis.

In summary, we have shown that WDR62 is involved in meiotic centriole duplication and manchette removal in addition to its established roles in meiotic initiation, meiotic spindle assembly and stability, all are important processes to generate functional spermatozoa for reproduction. Our results suggest that spermatocytes without normal centriole numbers most likely underwent prolonged meiotic metaphase that led to apoptosis. Round spermatids that inherited two centrioles continue to undergo spermiogenesis; however, WDR62 deficiency in elongated spermatids resulted in manchette removal defect, therefore causing oligoasthenoteratospermia and male infertility.

## Methods

**Animal husbandry and genotyping**. The Wdr62 genetrap Mus musculus (mice) on C57Bl/6 background were housed in the UQBR animal house facility. All animal experiments and breedings were approved by the Animal Ethics Committee at the University of Queensland. Both male and female adult heterozygous mice were used for timed matings. Mating was assumed at midnight and timed from 0.5 day. Embryos and newborns from both sexes were collected at E14.5, E16.5 and P0 for brain cortical analysis. Male mice at P5, P21, P28 and adult (>6 weeks old) were used for testis experiments, while female mice at 8 weeks old were used for ovary analysis. The genotype of each mouse or embryo was confirmed by PCR. DNA was extracted from the toe clips of mice or embryonic tails using the REDExtract-N-Amp Tissue PCR kit (Sigma) according to the manufacturer's instruction. The PCR reaction comprised of 1× REDExtract-N-Amp PCR reaction mix (Sigma), 0.4 µM Forward oligo P1 (5′ GTTGTGTCTGCTCTGTGTGG 3′), 0.4 µM Reverse oligo P2 (5′ CTGTGATGCCAAGCACC 3′), 0.4 µM Reverse oligo P3 (5′ CAACGGGTTCTTCTGTTAGTCC 3′), 2 µL DNA template and water to a final volume of 20 µL. The PCR cycling condition involved 35 cycles of 30 s denaturation at 94 °C, 30 s annealing at 60 °C and 45 s extension at 72 °C. The expected PCR size for wild type is one band at 485 bp, heterozygous two bands at 485 and 276 bp, and homozygous is one band at 276 bp.

**BrdU labelling**. Pregnant dams were injected intraperitoneally with 100 µL/10 g BrdU labelling reagent (Thermo Fisher Scientific) 24 h prior to embryo collection.

**Spermatocyte spreads and immunofluorescence analysis**. Spermatocyte spreads and immunofluorescence analysis were performed as described in Becherel et al.[62]. Briefly, testes were decapsulated, chopped and resuspended in DMEM (Gibco) before passing through a 100 µm cell strainer. Cells were pelleted by centrifuging at 500 r.p.m. at room temperature for 5 min and then resuspended in 0.1 M sucrose. Ten microliters of cell suspension was spread onto glass slides prewetted in 100% methanol or 4% paraformaldehyde and fixed for 2 h. The slides were subsequently washed with PBS and air-dried in the presence of Kodak Photo-Flo 200 (Kodak, 1 in 250 dilution). Slides were stored at −80 °C. For immunostaining, slides were rehydrated in PBS and blocked in blocking buffer (20% FBS, 0.2% Triton X in PBS) for 1 h before incubating with primary antibody at 4 °C overnight. The next day, slides were incubated with secondary antibody and counterstained with DAPI before mounting in Prolong Diamond (Invitrogen). The primary and secondary antibodies used are listed in Supplementary Table 1.

**Sperm motility assessment, smear, stain and counts**. Cauda epididymal spermatozoa were collected as described in Gu et al.[63]. Briefly, an incision was made to the caudal end of isolated epididymus in DMEM (Gibco). This was followed by 15 min incubation at 37 °C to allow spermatozoa to swim out. Ten microliters of sperm suspension was dropped into the centre grid of the hemocytometer and motile sperms were video captured using the Nikon eclipse TS100 microscope (×20 objective) with an iphone7 camera. The distance travelled and time taken of motile sperms were measured and graphed as speed (micrometre per second) in Graphpad Prism version 8.3. Sperms with head pinned on the hemocytometer were discarded from analysis. The remaining sperms were centrifugated at 150 g for 5 min to remove DMEM. Sperms were fixed in 2.5% glutaraldehyde with 0.1 M HEPES pH 7.5 overnight prior to counting on the hemocytometer as described in Wang[64]. Alternatively, sperms were fixed in 95% ethanol for 15 min and 10 µL was smeared onto a glass slide and air-dried for 2 h. Slides were kept at 4 °C overnight. The next day, sperm stain was performed using the Papanicolaou method as described in https://doi.org/10.1093/eshremonographs/2002.2.13, with 3 min incubation in Harris haematoxylin (Sigma), 2 min in orange G6 (OG6 0.5% solution in 95% ethanol, phosphotungstic acid 0.015 g/100 mL) and 5 min in EA50 (0.5% lightgreen SF yellowish CI 42095, 0.5% bismark brown CI 21000, 0.5% eosin Y CI 45380, 0.2 g phosphotungstic acid and 2 drops of lithium carbonate (0.154 g in 10 mL dH2O)).

**Transmission electron microscopy (TEM)**. TEM was performed according to Ngo et al.[65]. Briefly, sperms were fixed in 2.5% glutaraldehyde with 0.1 M HEPES pH 7.5 overnight. Subsequent processing was done using a Pelco Biowave (Ted Pella) processing microwave. The samples were post-fixed in 1% osmium tetroxide

prior to dehydration in increasing ethanol gradient and then in combination with increasing ratio of LR White Resin before embedding in pure LR White Resin and polymerised at 60 °C for 24 h. Ultrathin sections (60 nm) were made using an Ultracut UC6 ultramicrotome (Leica Microsystems) with a diamond knife then placed on copper grids. The grids were post-stained with 5% uranyl acetate in 50% ethanol for 2 min, followed by Reynolds lead citrate for 30 s at room temperature. Ultrathin sections were examined under a JEM-1011 (JEOL) or HT7700 (Hitachi) transmission electron microscope, both operated at 80 kV.

**Histology**. Brains and testes were fixed in 4% paraformaldehyde or Bouin's solution (Sigma) for at least 24 h. Samples were processed and microtome sectioned to 6 μm thickness. H&E staining was performed as described in Shohayeb et al.[25], with 4 min haematoxylin and 1 min eosin incubation. PAS-haematoxylin staining was performed according to the manufacturer's instructions (Sigma 395B-1KT) and staging of the seminiferous tubules was performed as described in Ahmed and de Rooij[41]. Slides were captured on an Olympus BX51 microscope using ×20 or ×60 objectives.

**qRT-PCR analysis**. RNA from testes and brains were isolated using PureLink RNA mini kit (Invitrogen) and cDNA synthesised using Superscript III (Invitrogen) according to the manufacturer's instructions. qRT-PCR was set up as previously described[66], consisting of 10 μL 2× SYBR green PCR mastermix (Applied Biosystems), 4 μL forward and reverse primers (concentration optimised), 4 μL cDNA (1 in 20 dilution) or cDNA without reverse transcriptase negative control (1 in 20 dilution) and water to a final volume of 20 μL. The primers for Sycp3, Rec8, Dmc1, Piwi1, Tdrd5, Tdrd6, Rfx2, Crem, Sox30, Tnp1 and Prm2 were published in Feng et al.[31]. The primers for Wdr62 and Mapkbp1 were Wdr62 Fwd 5′ACACAG AAGTCCCTACCCCA; Wdr62 Rev 5′CCAGCATGCGGTAAAGGTCA; Mapkbp1 Fwd 5′GGTACATGGCTCTACCCTGC; Mapkbp1 Rev 5′GCTCAGTGTCCAAC AAAGCC. qPCR was performed in triplicates for individual genes for each cDNA sample and data collected using QuantStudio 7 Flex system (ABI Systems). The cycle thresholds (CT) for each gene were determined in QuantStudio software version 1.3 (ABI Systems). The difference in mean cycle threshold (ΔCT) between the test gene and control gene, Gapdh or Tbp was determined. The conversion to expression values equal $2^{-ΔCT}$ assuming a doubling of PCR product in each round of amplification were analysed in Microsoft Excel 2016. The relative expression values were graphed using Graphpad Prism version 8.3.

**Immunoblotting and immunofluorescence**. Testes and brains were lysed in RIPA buffer (50 mM Tris-HCl pH 7.3, 150 mM NaCl, 0.1 mM EDTA, 1% sodium deoxycholate, 1% Triton X-100, 0.2% NaF and 100 μM $Na_3VO_4$ supplemented with protease inhibitors (Roche)) and quantitated using Bradford assay (BioRad) against BSA standard. Western blotting was performed as described in Lim et al.[9]. Briefly, 50 μg of protein lysates were ran on the 4–15% Tris-glycine gradient precast gel (BioRad), transferred to PVDF membrane (Millipore) and blocked in 5% skim milk, before incubating with primary antibody at 4 °C overnight. The next day, membrane was incubated with secondary antibody for 1 h and bands were detected using ECL Clarity (BioRad), visualised using Odyssey-Fc and bands were quantified using Image Studio Lite version 5.2. The primary and secondary antibodies used and their dilutions are listed in Supplementary Table 1.

Immunofluorescence on tissue sections were performed as described in Lim et al.[7] and Shohayeb et al.[25], with antigen retrieval performed by submerging rehydrated slides in 10 mM sodium tri-citrate pH 6.0 at 95 °C for 15 min. Whole mount immunofluorescence on P5 seminiferous tubules was performed as described in Hobbs et al.[67], with testes decapsulated and seminiferous tubules teased out before 4% paraformaldehyde fixation at 4 °C overnight. Both slides and seminiferous tubules were blocked in blocking buffer (20% FBS, 0.2% Triton X in PBS) for at least 1 h before incubating with primary antibody at 4 °C overnight. The next day, slides and seminiferous tubules were incubated with secondary antibody and counterstained with DAPI before mounting in Prolong Diamond (Invitrogen). Negative controls (using IgG antibody or no primary antibody) were included in each experiment. The primary and secondary antibodies as well as IgG control antibodies used and their dilutions are listed in Supplementary Table 1. Microscopy was performed using a Leica DMi8 SP8 inverted confocal microscope and Z-stacks were collected at 0.5 μm intervals. Z-stacked maximum intensity projection images were compiled using the Leica Application Suite Advanced Fluorescence Lite (LAS AF Lite) software version 2.6.3 build 8173 (Leica Microsystems).

TUNEL staining was performed using the fluorescein in situ cell death detection kit (Roche) according to the manufacturer's instructions.

**Statistics and reproducibility**. No sample size calculation was taken to predetermine sample size. At least three homozygous mice and at least three control littermates were analysed for each experiment to ensure reproducibility. All attempts at replication were successful; therefore, no data were excluded for analysis. The n number for each experiment is detailed in the figure legends. Each pair of male mice (homozygote and its littermate control) were randomly assigned for testes related experiments depending on availability and age. All experiments were performed in a blinded manner, with genotypes revealed at the graphing and/or figure compilation step. For brain cortex quantification, six separate areas of each

cortex were captured by confocal microscopy per section per brain. A 100 μm wide box were drawn for each Z-stacked image in Adobe Photoshop. All positively stained cells in the 100 μm wide box were counted manually. For centriole quantification, centrioles co-localised with Pericentrin in pachytene/diplotene spermatocytes were counted manually from Z-stacked images. Both pHH3 and TUNEL-positive tubules, γH2AX and SYCP3 co-stained spermatocytes were also manually counted from Z-stacked images. For CEP63 fluorescence intensity, original Z-stacked images were used, a constant region of interest (ROI) area was selected to quantify CEP63 and adjacent background intensities using the LAS AF Lite software version 2.6.3 (Leica Microsystems), each CEP63 ROI intensity were subtracted against the background intensity from the same image. All data were graphed as mean ± SEM and statistically analysed (two-tailed unpaired Student's t-test or two-way ANOVA) using Graphpad Prism version 8.3.

**Reporting summary**. Further information on research design is available in the Nature Research Reporting Summary linked to this article.

## Data availability
Data generated and analysed in this study are included in this article and its Supplementary Information files. Data of graphs in the main figures are provided as Supplementary Data 1/Source data for figures. Image datasets of the main figures are available on FigShare. Figure 1c https://doi.org/10.6084/m9.figshare.14451126; Fig. 2a https://doi.org/10.6084/m9.figshare.14456640.v1 and https://doi.org/10.6084/m9.figshare.14456571.v1; Fig. 2c https://doi.org/10.6084/m9.figshare.14458545.v1; Fig. 3a https://doi.org/10.6084/m9.figshare.14457006.v1; Fig. 3b https://doi.org/10.6084/m9.figshare.14457426.v1; Fig. 3c https://doi.org/10.6084/m9.figshare.14457435.v1; Fig. 4a https://doi.org/10.6084/m9.figshare.14457585; Fig. 4c https://doi.org/10.6084/m9.figshare.14457774.v1; Fig. 5a https://doi.org/10.6084/m9.figshare.14457852.v1; Fig. 5d https://doi.org/10.6084/m9.figshare.14457903.v1; and Fig. 6 https://doi.org/10.6084/m9.figshare.14457927.v1. All relevant data are available from the corresponding author on reasonable request.

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

## Acknowledgements

This project was funded by NHMRC (GNT1046032, GNT1162652), Australian Research Council (FT120100193) and Cancer Council Queensland (GNT1101931) grants awarded to A/Prof. D.C.H.N. We would like to thank Dr. Abrey Yeo and A/Prof. Mike Piper for reagents. We also acknowledge the facilities, and the scientific and technical assistance, of the Microscopy Australia Facility at the Centre for Microscopy and Microanalysis (CMM), Biological Resources/Animal housing (UQBR), SBMS Histology and SBMS Imaging facility, The University of Queensland.

## Author contributions

U.Y.H. and C.-W.A.F. designed the experiments; U.Y.H., Y.Y.Y., A.L.B., Z.W., M.E.R. and B.S. performed the experiments and analysis; U.Y.H., C.-W.A.F., H.H., K.K.K., J.B. and D.C.H.N. participated in discussion; U.Y.H. and D.C.H.N. wrote the manuscript.

## Competing interests

The authors declare no competing interests.
