## [Peer Review File · Communications Biology]

Reviewers' comments:

Reviewer #1 (Remarks to the Author):

In this manuscript Ho et al. generated a genetrapped mouse model deficient in Wdr62 to study the developmental effects of Wdr62 deficiency during meiosis in the testis. Wdr62 is a scaffold protein involved in centriole duplication and spindle assembly during mitosis and mutations in Wdr62 can cause primary microcephaly and premature ovarian insufficiency. The Author performed a lot of experiments and on the basis of the obtained data they suggested that proper Wdr62 function is necessary for timely spermatogenesis and spermiogenesis during male reproduction. The manuscript contains a great amount of experimental work supporting the ideas of the Authors and the observations reported here may give further insights to our knowledge about Wdr62 protein function.

The main concern this reviewer has is that sometimes the provided experiments may not be completely clear. In my opinion the paper is surely of interest and worthy of being published, but some revisions (especially in the Results) are necessary to make it accessible to a wider audience of readers.

Here my specific comments:

Introduction:

Page 3 lines 28-38: The Authors should summarize this initial part concerning meiosis.

Page 4 line 68: in my opinion the sentence "we have demonstrated that..." could be replaced with "our data indicate that..."

In the Introduction the Authors should stress more that MCPH genes influence the meiotic process.

Results:

Page 6 lines 105-107: this reviewer retains that it might be useful to better explain the importance of the data obtained on the Mapkbp1 expression.

Page 7 line 148: the Authors should explain because is done the Ddx4 labelling.

Page 8 line 162: the Authors should explain why is done the γ H2AX labelling.

Page 9 line 175: again, the Authors should explain why the labelling for Rad51 was performed.

Page 9 line 179: and again why they marked pHH3.

Page 10 line 200: the Authors should explain why they employed an anti-acetylated alpha tubulin antibody to mark spindle microtubules. Usually this antibody is utilized to mark stable microtubules (Axoneme).

Page 10 lines 212-218: the Authors should better explain this part of the results.

Page 11 line 238: the Authors may overlook the reference to the acrosome.

Page 13 line 279: the Authors should better explain CPAP and Kif3 b proteins

Figures

Fig. 4b: the picture shows a labelling for Centrin, however the text lacks the explanation of this experiment. Moreover, centrin is a centriolar protein, while in this picture it has a very different localization. The Authors should clarify.

Fig. 4e: the Authors should explain why they performed a double labelling for Aurka/pHH3.

Supplementary Fig.1: the Authors should add the letters a and c in the corresponding pictures.

Supplementary Table 1 Antibodies used in the study.

In this table the Authors reported the antibodies employed in immunoblotting and immunofluorescence experiments. In this list appears an anti-Cep170 and an anti-gamma tubulin antibody. However, this reviewer was not able to find any references to these antibodies in the text. The authors should check.

Anti-DDX4 / MVH antibody name should be appear entirely in the list.

Finally, control experiments should be reported in Materials and Methods section.

Reviewer #2 (Remarks to the Author):

WDR62 has previously been shown to be essential for both male and female fertility in the mouse. Specifically, during spermatogenesis prior studies have revealed WDR62 is essential for the timely initiation of germ cell meiosis during the first wave of spermatogenesis, and for proper formation of the meiotic spindle and that its loss results in spermatocytes arresting in metaphase I. The authors of the present study have further characterised the role of WDR62 in mouse spermatogenesis. They have beautifully characterised the initiation and progression of male meiosis in Wdr62 deficient mice, and consistent with the previous study, they show delays in meiotic initiation during the first wave of spermatogenesis, and once meiosis has initiated, metaphase arrest leading to apoptosis. Importantly, they reveal a novel role for WDR62 in spermatocytes centriole duplication and suggest the observed centriole under duplication in Wdr62 deficient spermatocytes underlies the metaphase defects. Finally, they also reveal novel roles for WDR62 spermatid head shaping via the manchette.

Overall the study provides new insights into the role of WDR62 during male fertility, and the authors have begun to investigate potential mechanism of actions of WDR62 in the testis via binding partners and changes in localisation of key proteins involved in meiosis and spermatid remodelling. However, the characterisation of the spermatogenesis phenotype needs greater attention to fully define the role of WDR62 in the testis and to clarify the authors the findings.

Specific points:

- Throughout the introduction of when discussing the role of various proteins in meiosis should always clarify whether you are referring to male vs female meiosis as the two processes are not entirely analogous. Related to that point on lines 32-33, 'meiosis' should be changed to male meiosis as across most species female meiosis is acentriolar.
- Throughout need to ensure have gene/protein nomenclature correct, when referring to mouse protein WDR62 should be capitalised.
- For the breeding assays were mice plug checked i.e. to ensure that lack of pups was not due to changes in mating behaviour? Also were control breeding pairs set up?
- Fig 1 panel A – there is a lot of fat still associated with the testes pictured was this removed prior to recording of weights?
- The authors report there were less round spermatids and elongating spermatids in the seminiferous epithelium of the Wdr62 deficient, which is consistent with the significant increase in spermatocytes undergoing apoptosis. However, on line 139 they report there was no significant difference in the expression of early round spermatid genes suggesting the numbers of round spermatids is normal. Does this mean the meiotic defect does not significantly impact on numbers of spermatids entering spermiogenesis?
- Related to the point above the authors state on line 141 there is a decrease in expression of TNP1 and PRM2 which could be due to a decrease in the numbers in elongating spermatids or due to a lack of compaction of the haploid genome. Precise analysis of elongated spermatid numbers in the testes (e.g. testis sperm production counts) would help clarify this, combined with more precise analysis of spermatid nuclear condensation. Given how few sperm reach the epididymis it is likely the former (i.e. less spermatids) which then raises the question of when during spermiogenesis are spermatids being lost and why – more thorough characterisation and presentation of the testis PAS histological data would help.
- On line 181 the authors state that AurkB levels were reduced in the Wdr62 mutant testes saying this suggests that anaphase was disturbed. Were anaphase defects observed in Wdr62 mutant testis sections or isolated cells, and what specific defects would you expect?
- Line 215 – should not report on data that isn't shown.
- The manchette phenotype is interesting and is a novel finding in terms of WDR62 function however much more care needs to be taken the characterisation of this phenotype.
 - o Staining of testis sections for alpha or acetylated tubulin is needed to validate the step-specific manchette phenotypes presented in Fig 6 and S Fig 2, as given the severe abnormalities of the Wdr62 mutant spermatids manchettes it's possible they differ from expectations.
 - o Line 252 – most data now suggests the manchette does not nucleate from either the centrosome or the perinuclear ring data, instead it is thought to nucleates from a region known as the centriolar adjunct – refer to papers from Lehti and colleagues.
 - o Presenting single channels as well as the merged channel for Fig 6b-d would allow better assessment of the localisation of each protein
 - o The katanin p60 staining looks very unspecific (do you have a negative control for this?), and of note a different katanin protein, KATNAL2 has been shown to be the mediator of katanin p80 function at the manchette.
 - o The authors conclude that accumulation of Cep63 at the centriole drives the manchette phenotype but present no data showing the two phenotypes are linked. Need to be careful with this conclusion.

Indeed, centriole defects are often seen in parallel with manchette defects but normally due to independent functions of the mutated proteins i.e. it functions both at the manchette and the centriole.

- Despite being infertile the WDR62 deficient mice still produce some sperm and these sperm have tails, as such, the fertilisation capacity/function of these sperm should be assessed e.g. through motility assessments. Indeed, given that WDR62 is a centrosome associated protein and you observed abnormal accumulation of CEP63 at the basal body, defects in flagella function are highly possible.

- Related to the above point while the authors have nicely shown that the sperm axoneme still forms via acetylated tubulin staining, light microscopy of HnE stained epididymal sperm is needed to assess the overall morphology of the sperm and flagella (the current stain does not reveal anything about accessory structures for example, whose formation can often be affected in mutants of centriole and microtubule associated proteins). Moreover, ultrastructural analysis is really needed before the authors can conclude whether the axoneme and HTCA is normal.

- The immunofluorescence data present throughout the paper is integral for many of the authors conclusions. However, no negative controls for any of the assays have been included to verify the specificity of the staining/fluorescence signal.

10th March 2021

Dear Reviewers,

We thank you for your valuable comments. We have addressed your comments and incorporated the new data into the revised manuscript entitled “WDR62 is required for centriole duplication in spermatogenesis and manchette removal in spermiogenesis”.

WDR62 is a centrosome associated protein that plays a vital role in centriole duplication and spindle assembly and stability in mitosis. In this paper, we have used a *Wdr62* genetrapped/deficient mouse model to show that WDR62 is required for efficient centriole duplication during meiosis and manchette removal in shaping the sperm head.

We have re-analysed our PAS-H staining of adult testis sections and further characterised the cauda epididymal spermatozoa through sperm counts, sperm motility, morphology and ultrastructural analyses. We have shown that WDR62 deficiency leads to reduced number of round spermatids entering spermiogenesis. The cauda epididymal spermatozoa showed reduced numbers, reduced motility and misshapen sperm heads, characteristics of a condition called oligoasthenoteratospermia or OAT, a common cause of subfertility in men. Furthermore, we have included our negative controls of our immunofluorescence experiments as well as single channel images as requested. To accommodate for all these changes, some of our original figures were rearranged. These changes are outlined in the table at the end of the reviewers' comments. The results obtained in this paper are significant because they provide additional insight into the previously known role of WDR62 in meiosis and microtubule disassembly. This paper should be of interest to readers in the areas of centrosome biology and molecular genetics of sperm development.

Please address all correspondence concerning this manuscript to me at u.ho@uq.edu.au.

Thank you for reviewing this manuscript.

Yours sincerely,

Dr. Uda Ho
Post-doctoral research fellow
School of Biomedical Sciences
University of Queensland
Australia

Reviewers' comments:

Reviewer #1 (Remarks to the Author):

In this manuscript Ho et al. generated a genetrapp mouse model deficient in Wdr62 to study the developmental effects of Wdr62 deficiency during meiosis in the testis. Wdr62 is a scaffold protein involved in centriole duplication and spindle assembly during mitosis and mutations in Wdr62 can cause primary microcephaly and premature ovarian insufficiency. The Author performed a lot of experiments and on the basis of the obtained data they suggested that proper Wdr62 function is necessary for timely spermatogenesis and spermiogenesis during male reproduction.

The manuscript contains a great amount of experimental work supporting the ideas of the Authors and the observations reported here may give further insights to our knowledge about Wdr62 protein function.

The main concern this reviewer has is that sometimes the provided experiments may not be completely clear. In my opinion the paper is surely of interest and worthy of being published, but some revisions (especially in the Results) are necessary to make it accessible to a wider audience of readers.

Here my specific comments:

Introduction:

1) Page 3 lines 28-38: The Authors should summarize this initial part concerning meiosis. Thank you for your suggestion. We have summarised the first paragraph of our Introduction concerning meiosis (Page 3 lines 31-36):

Meiosis is the process where a diploid cell divides twice to produce four haploid gametes¹, whereas a diploid cell divides once in mitosis to generate two diploid cells². The centrosome is the main microtubule organising centre (MTOC) consisting of one pair of centrioles surrounded by pericentriolar matrix (PCM) that anchors the microtubules. The centrioles are replicated alongside DNA during S-phase and in metaphase they form the spindle poles³. However in male meiosis II, the centrioles are predicted to duplicate in the absence of DNA replication⁴.

2) Page 4 line 68: in my opinion the sentence "we have demonstrated that..." could be replaced with "our data indicate that..."

Page 4 line 68, now page 4 line 67, has been replaced with "Our data indicated that..."

3) In the Introduction the Authors should stress more that MCPH genes influence the meiotic process.

We have summarised the current literature regarding the various roles of each MCPH gene in meiosis in our Introduction on page 3-4, lines 53-60:

In addition to neurological defects, these genes are also implicated in meiosis, including meiotic initiation, homologous recombination and spermatocyte cell divisions to form round spermatids. Microcephelin (MCPH1) is required for homologous pairing of

chromosomes, meiotic recombination DNA repair and maintaining genome stability in mice. CDK5RAP2 (also known as MCPH3) and ASPM (also known as MCPH5) are required for germ cell maintenance. CEP152 (also known as MCPH9) is involved in meiotic spindle formation in the mouse oocyte. CENP-E (also known as MCPH13) is localised to the kinetochores during mitosis and male meiosis.

Results:

4) Page 6 lines 105-107: this reviewer retains that it might be useful to better explain the importance of the data obtained on the *Mapkbp1* expression.

MAPKBP1 is a paralog and binding partner of WDR62. It is also expressed in the testis and shares similar cellular localisation and interacting partners with WDR62, thus can potentially compensate for WDR62 loss. We used qRT-PCR to examine if *Mapkbp1* was upregulated to compensate for WDR62 deficiency. However, similar *Mapkbp1* mRNA expression were detected in *Wdr62^{tm1a/tm1a}* and littermate control P0 brains (Supplementary Figure 1q), indicating that the expression of *Mapkbp1* is not increased to compensate for WDR62 deficiency. These information can be found in page 5-6 lines 104-109 of our manuscript.

5) Page 7 line 148: the Authors should explain because is done the Ddx4 labelling.

We have now explained what DDX4 staining marks when it first appeared in page 6 line 121-122:

"..staining for the germ cell marker DEAD-Box helicase 4 (DDX4, also known as mouse VASA homolog, MVH)".

We have performed DDX4 (marks germ cells) and STRA8 (marks meiotic initiation) co-whole mount staining of P5 seminiferous tubules to show the presence of meiotic initiation in germ cells (page 8 lines 156-158):

Next, we examined if meiotic initiation was delayed in our *Wdr62^{tm1a/tm1a}* testis. We used whole mount staining and quantified the number of DDX4+ germ cells co-expressing STRA8, the latter indicating that meiosis has initiated.

6) Page 8 line 162: the Authors should explain why is done the γ H2AX labelling.

γ H2AX marks double stranded DNA breaks and the sex chromosomes. This information has been added to page 8 line 172-173. Together with SYCP3, which marks the synaptonemal complex that aligns sister chromatids, their staining patterns can be used to distinguish the sub-phases of prophase I. Zygotene is characterised by spaghetti-like SYCP3 staining with a cloud of γ H2AX staining over the nucleus, as sister chromatids started to pair and double stranded DNA breaks were presented in preparation for homologous recombination. Pachytene is characterised by short, intense and thick SYCP3 staining, as sister chromatids align and recombine, while γ H2AX marks the sex chromosomes. In diplotene, the ends of SYCP3 stained sister chromatids began to unpaired, while γ H2AX continues to mark the sex chromosomes. This information regarding the staining patterns of SYCP3 and γ H2AX in distinguishing the sub-phases of prophase I has been added to page 8 lines 175-180.

7) Page 9 line 175: again, the Authors should explain why the labelling for Rad51 was performed.

RAD51 is involved in double stranded DNA damage repair after homologous recombination. It is also expressed in meiosis during prophase I performing the same function.

The following information has been added to page 9 lines 193-194:

Prophase I was marked by γ H2AX and RAD51, with the latter involved in double stranded DNA damage repair after homologous recombination.

8) Page 9 line 179: and again why they marked pHH3.

We have used pHH3 to mark late G2/M phase of the cell cycle, and an increase in pHH3 indicates more cells in G2/M phase or G2/M arrest.

The following information has been added to page 9 lines 198-199:

However, phospho-Histone H3 Ser10 (pHH3), a marker for condensed chromosomes during late G2 and metaphase, was substantially increased in *Wdr62^{tm1a/tm1a}* P28 and adult testis suggesting possible prolonged metaphase or G2/M arrest (Figure 2d).

9) Page 10 line 200: the Authors should explain why they employed an anti-acetylated alfa tubulin antibody to mark spindle microtubules. Usually this antibody is utilized to mark stable microtubules (Axoneme).

We apologise for this error. The images in original Figure 4c were in fact anti- α tubulin staining and not anti-acetylated α -tubulin as labelled. This has been corrected in the relevant sentence (page 10 line 223) and figure (Supplementary Figure 3b).

The spindle microtubules are also acetylated, and therefore acetylated α -tubulin staining would also have shown the position and alignment of meiotic spindles. Analysis of mitotic spindles using acetylated α -tubulin staining has been used in Tomoaki Nagai, Masanori Ikeda, Shuhei Chiba, Shin-ichiro Kanno, Kensaku Mizuno; *Journal of Cell Science* 2013 126: 4369-4380; doi: 10.1242/jcs.127209.

10) Page 10 lines 212-218: the Authors should better explain this part of the results.

We believed the reviewer is seeking a better explanation of why we have examined spindle assembly checkpoint (SAC). SAC can be activated to prevent the onset of anaphase until the chromosomes are all aligned and can trigger apoptosis if the chromosomes failed to align. Based on the study by Qin et al., 2019, we were anticipating that the loss of WDR62 would cause spindle assembly defect, which would lead to metaphase I arrest and SAC activation in the germ cells of *Wdr62* deficient testis. We have added the following information in page 10 lines 221-223 to clarify why we examined the meiotic spindles:

Since conditional *Wdr62* knockout in male meiosis has previously reported to show spindle assembly defect and spindle assembly checkpoint (SAC) activation, we first performed an analysis of meiotic spindles using α -tubulin staining,...

However, we have failed to detect SAC activation (via Western analysis of SAC activation markers e.g. MAD2, BUBR1, BUB3, CENPE) despite a significant increase in pHH3 and TUNEL positive germ cells in our WDR62 deficient model (Figure 3c). As the BUBR1, BUB3 and CENPE western analyses gave negative results (blank blots) and we were asked by reviewer 2 to remove data that is not shown, the mentioning of these SAC markers were removed from our manuscript. We have decided to leave only the description of the MAD2 western, representing our attempt to detect SAC activation, in page 11 lines 231-234:

We also examined the expression of MAD2, a spindle assembly checkpoint component that prevents the onset of anaphase until chromosomes were aligned, but no difference was observed between *Wdr62^{tm1a/tm1a}* and control adult testis (Supplementary Figure 3e). Therefore, spindle misalignment observed in WDR62 deficient testis was not due to spindle assembly defect, but rather spindle disorientation.

11) Page 11 line 238: the Authors may overlook the reference to the acrosome. The reference, Ahmed, E. A. & de Rooij, D. G. Staging of mouse seminiferous tubule cross-sections. *Methods Mol Biol* **558**, 263-277, doi:10.1007/978-1-60761-103-5_16 (2009), was added to page 11 line 255 reference #41.

12) Page 13 line 279: the Authors should better explain CPAP and Kif3 b proteins CPAP and KIF3 β both marks cilium/flagellum elongation.

The following information was added to page 12 lines 274-278:

We have recently reported the involvement of WDR62 in primary cilium formation, thus WDR62 deficiency may also affect proper flagellum formation. We performed acetylated α -tubulin immunofluorescence on adult spermatocyte spreads, western analysis of cilium/flagellum elongation markers, CPAP and KIF3 β , and transmission electron microscopy to examine flagellum formation in WDR62 deficient sperms.

Figures

13) Fig. 4b: the picture shows a labelling for Centrin, however the text lacks the explanation of this experiment. Moreover, centrin is a centriolar protein, while in this picture it has a very different localization. The Authors should clarify. We thank the reviewer for highlighting this. WDR62 is localised to the proximal centrioles during S-phase (Kodani et al., 2015), and mitotic spindles and spindle poles during M-phase in mitosis (Lim et al., 2015). Thus we anticipated that WDR62 would show similar localisation during meiosis. When we performed this experiment, the Qin et al., 2019 paper was not published yet (who generated a HA-tagged *Wdr62* mouse to show WDR62 localisation to the spindles during meiosis). Therefore, we decided to use Centrin to mark the centrioles, anticipating that WDR62 would localise to the proximal centrioles during S-phase/prophase I, and meiotic spindles and spindle poles during metaphase. We don't know why the mouse anti-Centrin antibody (Millipore 04-1624) show non-specific staining on the spindles of mouse metaphase spermatocytes. Since our focus in

this result section is on WDR62 localisation during metaphase, to avoid confusion, we have removed the Centrin stain (red channel), leaving the WDR62 (green channel) and merge with DAPI (blue). We believe the images shown in Supplementary Figure 3b still support WDR62 localisation to the meiotic spindles during metaphase in *Wdr62^{+/+}* and absent in *Wdr62^{tm1a/tm1a}*.

14) Fig. 4e: the Authors should explain why they performed a double labelling for AurkA/pHH3.

We have performed double labelling for AURKA/pHH3 to examine the localisation of AURKA during prometaphase or metaphase despite no change was observed in AURKA protein expression by western blotting. WDR62 has been previously shown to regulate AURKA localisation and AURKA is significant in WDR62 spindle localisation (Lim et al., 2016). Unfortunately, no difference in AURKA localisation was observed between *Wdr62^{+/+}* and *Wdr62^{tm1a/tm1a}*. As it is a negative result and in order to address other reviewer comments, we have now decided to omit this experiment.

15) Supplementary Fig.1: the Authors should add the letters a and c in the corresponding pictures.

We thank the reviewer for picking up on these errors. We have now added the missing letters to the appropriate panels in Supplementary Figure 1 in our revised manuscript.

Supplementary Table 1 Antibodies used in the study.

16) In this table the Authors reported the antibodies employed in immunoblotting and immunofluorescence experiments. In this list appears an anti-Cep170 and an anti-gamma tubulin antibody. However, this reviewer was not able to find any references to these antibodies in the text. The authors should check.

We thank the reviewer for picking up this discrepancy. Mention of these antibodies were in reference to experimental data in previous versions of the manuscript that were removed in the final version for clarity. We have now removed mention of Anti-Cep170 and gamma tubulin antibodies from Supplementary Table 1.

17) Anti-DDX4 / MVH antibody name should be appear entirely in the list.

The full DDX4/MVH antibody name is detailed in Supplementary Table 1.

18) Finally, control experiments should be reported in Materials and Methods section. IgG control images of the immunofluorescence experiments in the main figures (Figure 2a, 3a, 4a, 4c and 6a) are now in Supplementary Figure 7. Negative controls for the supplementary figures are in their relevant figures. Immunofluorescence and control experiments were performed at the same time. The IgG antibodies used are recorded in Supplementary Table 1.

Reviewer #2 (Remarks to the Author):

WDR62 has previously been shown to be essential for both male and female fertility in

the mouse. Specifically, during spermatogenesis prior studies have revealed WDR62 is essential for the timely initiation of germ cell meiosis during the first wave of spermatogenesis, and for proper formation of the meiotic spindle and that its loss results in spermatocytes arresting in metaphase I. The authors of the present study have further characterised the role of WDR62 in mouse spermatogenesis. They have beautifully characterised the initiation and progression of male meiosis in *Wdr62* deficient mice, and consistent with the previous study, they show delays in meiotic initiation during the first wave of spermatogenesis, and once meiosis has initiated, metaphase arrest leading to apoptosis. Importantly, they reveal a novel role for WDR62 in spermatocytes centriole duplication and suggest the observed centriole under duplication in *Wdr62* deficient spermatocytes underlies the metaphase defects. Finally, they also reveal novel roles for WDR62 spermatid head shaping via the manchette.

Overall the study provides new insights into the role of WDR62 during male fertility, and the authors have begun to investigate potential mechanism of actions of WDR62 in the testis via binding partners and changes in localisation of key proteins involved in meiosis and spermatid remodelling. However, the characterisation of the spermatogenesis phenotype needs greater attention to fully define the role of WDR62 in the testis and to clarify the authors the findings.

Specific points:

19) Throughout the introduction of when discussing the role of various proteins in meiosis should always clarify whether you are referring to male vs female meiosis as the two processes are not entirely analogous. Related to that point on lines 32-33, 'meiosis' should be changed to male meiosis as across most species female meiosis is acentriolar. We thank reviewer 2 for this excellent suggestion. We have now made the following changes to lines 35, 60 and 61 on page 3-4 to more specifically delineate male meiosis from female meiosis.

We have also added the ovary data (page 6 lines 120-125; see also reviewer comment 21) and we used words like female and ovary to carefully indicate female meiosis in this context.

20) Throughout need to ensure have gene/protein nomenclature correct, when referring to mouse protein WDR62 should be capitalised.

All mouse protein mentioned in immunofluorescence and western blotting experiments and the names of antibodies used are now capitalised in this paper. All mouse mRNA mentioned in qRT-PCR experiments have the first letter in capital and non-italicised. All mouse gene names have the first letter in capital and are italicised. We hope this will satisfy the reviewer request and journal guidelines.

21) For the breeding assays were mice plug checked i.e. to ensure that lack of pups was not due to changes in mating behaviour? Also were control breeding pairs set up? Eight sets of *Wdr62*^{+/+} x *Wdr62*^{+/tm1a} control breeding pairs were set up. This information has been added to page 6 line 115.

Plugs were not checked, however, we have shown that *Wdr62*^{tm1a/tm1a} female mice are also sterile by performing DDX4 staining in adult ovary sections, which marks germ cells. The following information was added to page 6 lines 120-125 and Supplementary Figure 2a and b:

Therefore, we investigated if ovary development was impaired in our *Wdr62*^{tm1a/tm1a} female mice by staining for the germ cell marker DEAD-Box helicase 4 (DDX4, also known as mouse VASA homolog, MVH). *Wdr62*^{+/+} 8 week old ovaries showed DDX4+ germ cells, however, smaller ovaries without DDX4+ germ cells were observed in *Wdr62*^{tm1a/tm1a} littermates (Supplementary Figure 2a and b), indicating an absence of oocytes. This result supports that WDR62 deficient females are unable to reproduce.

22) Fig 1 panel A – there is a lot of fat still associated with the testes pictured was this removed prior to recording of weights?

We can clarify that testes were photographed as soon as they were isolated. Fat was removed from all the testes prior to measuring their weights. The P28 data have now moved to Supplementary Figure 2c to 2e as space is required in Figure 1 for the PAS-H data that we have re-analysed as requested in comment 24.

23) The authors report there were less round spermatids and elongating spermatids in the seminiferous epithelium of the *Wdr62* deficient, which is consistent with the significant increase in spermatocytes undergoing apoptosis. However, on line 139 they report there was no significant difference in the expression of early round spermatid genes suggesting the numbers of round spermatids is normal. Does this mean the meiotic defect does not significantly impact on numbers of spermatids entering spermiogenesis?

Meiotic defect did impact on the number of spermatids entering spermiogenesis. From our PAS-H analysis (please see reviewer comment 24), we have observed reduced number of round spermatids as early as in stage III seminiferous tubules (Figure 1e). No difference are detected in the expression of early round spermatid genes from our qRT-PCR analysis is because the mRNA transcripts of *Rfx2*, *Crem* and *Sox30* have previously reported to be expressed at the onset of meiosis, but are not translated till the germ cells reach spermiogenesis (Delmas et al., 1993; Horvath et al., 2004; Zhang et al., 2018).

The following information was added to page 7 lines 145-149:

Early round spermatids express genes encoding *Rfx2*, *Crem* and *Sox30* transcription factors that are essential for the progression of spermiogenesis, we did not observe differences in the mRNA expression of these genes between *Wdr62*^{+/+} and *Wdr62*^{tm1a/tm1a} P28 and adult testes, most likely because mRNA transcripts of these genes are produced early in meiosis but are not translated till spermiogenesis.

24) Related to the point above the authors state on line 141 there is a decrease in expression of TNP1 and PRM2 which could be due to a decrease in the numbers in elongating spermatids or due to a lack of compaction of the haploid genome. Precise analysis of elongated spermatid numbers in the testes (e.g. testis sperm production counts) would help clarify this, combined with more precise analysis of spermatid nuclear condensation. Given how few sperm reach the epididymis it is likely the former (i.e. less spermatids) which then raises the question of when during spermiogenesis are spermatids being lost and why – more thorough characterisation and presentation of the testis PAS histological data would help.

We thank the reviewer for their suggestion. Taking these comments on board, we have re-analysed the PAS-H data, counted the number of spermatocytes, round and elongated spermatids in stage III, VII and XII seminiferous tubules, and found similar number of spermatocytes and spermatocytes in metaphase between *Wdr62*^{+/+} and *Wdr62*^{tm1a/tm1a}. However, we have shown a loss of round and elongated spermatids in *Wdr62*^{tm1a/tm1a} stage III, VII and XII seminiferous tubules compared to control littermates (Figure 1e), suggesting that spermatids were lost as they enter spermiogenesis.

The following information has been added to page 6-7 lines 129-137 and the new data were added to Figure 1d to e:

Periodic acid Schiff-hematoxylin (PAS-H) staining of adult testis cross-sections revealed approximately 2.4 fold increase in the percentage of Stage XII seminiferous tubules (Figure 1d). Further examination of the number of spermatocytes and spermatids at stage III, VII and XII seminiferous tubules showed similar number of primary spermatocytes and spermatocytes in metaphase between *Wdr62*^{+/+} and *Wdr62*^{tm1a/tm1a} (Figure 1e). However, a loss of round spermatids (61% reduction at stage III and 73% reduction at stage VII) and elongated spermatids (71% reduction at stage III, 82% reduction at stage VII and 79% reduction at stage XII) were observed in *Wdr62*^{tm1a/tm1a} seminiferous tubules (Figure 1e), suggesting that spermatids were lost during early spermiogenesis.

25) On line 181 the authors state that AurkB levels were reduced in the *Wdr62* mutant testes saying this suggests that anaphase was disturbed. Were anaphase defects observed in *Wdr62* mutant testis sections or isolated cells, and what specific defects would you expect?

Based on the study by Qin et al., 2019, we were anticipating metaphase I arrest and spindle assembly checkpoint (SAC) activation in the germ cells of *WDR62* deficient testis. However, we have failed to detect SAC activation (via Western analysis of SAC activation markers e.g. MAD2, BUBR1, BUB3, CENPE) despite a significant increase in pHH3 and TUNEL positive germ cells in our *WDR62* deficient model (Figure 3c). Since we have observed reduced number of round and elongated spermatids in our *Wdr62*^{tm1a/tm1a} seminiferous tubules (Figure 1e), it is most likely that reduced AURKB protein level in the *Wdr62*^{tm1a/tm1a} adult testis (Figure 2d) is due to less cells that underwent anaphase. This information has been added to page 9 lines 200-204:

We also examined the metaphase to anaphase transition marker, Aurora Kinase B (AURKB), which showed reduced AURKB protein level in *Wdr62*^{tm1a/tm1a} adult testis (Figure 2d),

indicating a decrease in spermatocytes undergoing anaphase. This result correlates well with our observation of reduced numbers of round spermatids in WDR62 deficient testis (Figure 1e).

26) Line 215 – should not report on data that isn't shown.

The sentence related to BUBR1, BUB3 and CENP-E are removed as western analysis of these proteins gave negative results and hence we couldn't show these data.

27) The manchette phenotype is interesting and is a novel finding in terms of WDR62 function however much more care needs to be taken the characterisation of this phenotype.

o Staining of testis sections for alpha or acetylated tubulin is needed to validate the step-specific manchette phenotypes presented in Fig 6 and S Fig 2, as given the severe abnormalities of the *Wdr62* mutant spermatids manchettes it's possible they differ from expectations.

Unfortunately, we do not have an antibody against the acrosome to assist in staging the seminiferous tubules/spermatids. But we did performed α -tubulin immunofluorescence on adult testis sections and examined the manchette of elongated spermatids in stage XII seminiferous tubules (characterised by spermatocytes in metaphase). In *Wdr62*^{+/+}, manchette encapsulating the caudal side of the sperm head could be seen; whereas in *Wdr62*^{tm1a/tm1a}, both sperm head and manchette were elongated comparatively (Supplementary Figure 5b). The phenotypes of these WDR62 deficient spermatids were similar to the acetylated α -tubulin immunofluorescence performed on our spermatocyte spreads, in particular step 12 spermatids (Supplementary Figure 5a). We hope this will be satisfactory for reviewer 2.

This information has been added to page 13 lines 298-300:

We confirmed this observation by performing α -tubulin staining in adult testis sections which showed misshapen sperm head and elongated manchette in *Wdr62*^{tm1a/tm1a} step 12 spermatids compared to *Wdr62*^{+/+} (Supplementary Figure 5b).

28) Line 252 – most data now suggests the manchette does not nucleate from either the centrosome or the perinuclear ring data, instead it is though to nucleates from a region known as the centriolar adjunct – refer to papers from Lehti and colleagues.

We appreciate reviewer 2 highlighting this specific distinction and suggested correction. In response we have replaced the original phrase "possibly nucleating from the perinuclear ring or the centrosome" in line 252 with "possibly nucleating from the centriolar adjunct associated with the proximal centriole" (Lehti and Sironen, 2016) in page 13 lines 289-290.

29) Presenting single channels as well as the merged channel for Fig 6b-d would allow better assessment of the localisation of each protein

The single channels for original Figure 6b, c and d are now added. Due to space limitations, original Figure 6b is now Supplementary Figure 5d, whereas original Figure 6c and d are now Figure 6 a and b respectively.

30) The katanin p60 staining looks very unspecific (do you have a negative control for this?), and of note a different katanin protein, KATNAL2 has been shown to be the mediator of katanin p80 function at the manchette.

Katanin p60 (KATNA1) and Katanin p80 (KATNB1) staining were performed at the same time and the IgG negative control for that experiment has been included in Figure 6a. Unfortunately we do not have a siRNA for Katanin p60 or tissue sections from Katanin p60 knocked out animals to show the non-specificity of the Katanin p60 antibody. We hope that the rabbit or mouse IgG control antibodies used in our negative controls are sufficient to support antibody specificity.

Thank you for your suggestion to examine KATNAL2 in our *Wdr62^{tm1a/tm1a}* and control spermatocyte spreads or testis sections, however, we are unable to obtain a commercial KATNAL2 antibody for our experiments. The KATNAL2 N-terminus antibody (SC-84855) from Santa Cruz Biotechnology used in Dunleavy JEM, Okuda H, O'Connor AE, Merriner DJ, O'Donnell L, Jamsai D, et al. (2017) Katanin-like 2 (KATNAL2) functions in multiple aspects of haploid male germ cell development in the mouse. PLoS Genet 13(11): e1007078 is no longer available. We have ordered a rabbit anti-KATNAL2 antibody (MBS416764) from MyBioSource/JL Research as it was our intention to include this additional Katanin protein in our study as suggested. However we are experiencing long delays as the supplier has informed us that longer production time is required. We have therefore decided to resubmit our revision. Whilst we agree that KATNAL2 stain could additional support our findings, we believed that our conclusion of *Wdr62* deficiency leads to reduced or delayed Katanin p80 accumulation in the manchette and manchette removal defect is appropriately supported by our existing data with Katanin p60 and p80 stains. We hope this will be satisfactory for reviewer 2.

31) The authors conclude that accumulation of Cep63 at the centriole drives the manchette phenotype but present no data showing the two phenotypes are linked. Need to be careful with this conclusion. Indeed, centriole defects are often seen in parallel with manchette defects but normally due to independent functions of the mutated proteins i.e. it functions both at the manchette and the centriole.

We appreciate reviewer 2 comments and agree that the two phenotypes may not be necessarily linked. We have revisited and removed mention in the manuscript that directly attributes CEP63 accumulation at the centriole drives the manchette phenotype (page 2 line 22) and modified other statements so that the statements are not so strong. Original lines 266-267 "...possibly leading to timely manchette removal and shaping of the sperm head" has been replaced with "...but how CEP63 is involved in this process is still unclear." in page 14 line 307 and also original lines 328-330 "...which appeared to be necessary for timely manchette removal (Figure 6b)" has been replaced with "...however, the exact role of CEP63 in spermiogenesis remains unknown." in page 16 lines 370-371. We have also moved the original Figure 6b (CEP63 localisation in elongated spermatids) to Supplementary Figure 5d to reduce the emphasis of this conclusion, as well as putting the cauda epididymal spermatozoa characterisation in between the centriole biogenesis

and manchette characterisation results to separate these two sections. In addition, we have acknowledged this reviewer's comment in our Discussion (page 16 lines 364-369): Our data leaves open to the possibility that the centriole defect and the manchette defect observed in our WDR62 deficient testis are independent events, as centriole biogenesis and manchette removal occur at different times and stages during sperm development. Although these centrosome associated proteins may be localised to both the centrioles and manchette, they may have different functions which require further investigations.

32) Despite being infertile the WDR62 deficient mice still produce some sperm and these sperm have tails, as such, the fertilisation capacity/function of these sperm should be assessed e.g. through motility assessments. Indeed, given that WDR62 is a centrosome associated protein and you observed abnormal accumulation of CEP63 at the basal body, defects in flagella function are highly possible.

Thank you for the suggestion. We have performed sperm motility assessments and added these data into our manuscript on page 12 lines 269-272 and Figure 5c. Our cauda epididymal sperm motility assessment measured the progressive movement of sperms, which showed approximately 50% reduction in motility of *Wdr62^{tm1a/tm1a}* sperms (Figure 5c). This alone may decrease the chance of reaching the egg for fertilisation, but not completely compromising the ability of fertilisation in WDR62 deficient sperms. We believe that reduced sperm motility in combination with low sperm counts (Figure 5b) causes infertility in WDR62 deficient male mice.

33) Related to the above point while the authors have nicely shown that the sperm axoneme still forms via acetylated tubulin staining, light microscopy of HnE stained epididymal sperm is needed to assess the overall morphology of the sperm and flagella (the current stain does not reveal anything about accessory structures for example, whose formation can often be affected in mutants of centriole and microtubule associated proteins). Moreover, ultrastructural analysis is really needed before the authors can conclude whether the axoneme and HTCA is normal.

We further examined the head-tail coupling apparatus (HTCA), axoneme and flagellum of our *Wdr62* deficient spermatozoa by performing sperm smear and staining as well as transmission electron microscopy (TEM), which showed normal axoneme and flagellum (Figure 5d and Supplementary Figure 4d). However, the HTCA of WDR62 deficient sperm appeared hollow (Supplementary Figure 4d) and this might account for the poor motility observed (Figure 5c).

We have added a section on *Wdr62^{tm1a/tm1a}* spermatozoa and oligoasthenoteratospermia in page 12-13 lines 264-284 and relevant Figure 5b (sperm counts), 5c (sperm motility), 5d (sperm staining on morphology) and Supplementary Figure 4d (TEM data).

34) The immunofluorescence data present throughout the paper is integral for many of the authors conclusions. However, no negative controls for any of the assays have been included to verify the specificity of the staining/fluorescence signal.

As per reviewer 1, the IgG control images of the immunofluorescence experiments in the main figures (Figure 2a, 3a, 4a, 4c and 6a) are now in Supplementary Figure 7.

Immunofluorescence and control experiments were performed at the same time. The IgG antibodies used are recorded in Supplementary Table 1.

Updated figures are summarised below:

New Figures are added to address reviewers' comments		
Figure number	Reviewer comments number	Details
Figure 1e	24	Stages III, VII and XII number of spermatocytes vs round spermatids vs elongated spermatids
Figure 5b	33	Sperm counts
Figure 5c	33	Sperm motility
Figure 5d	33	Sperm stain for morphology
Supplementary Figure 2a	21	Ovary photo
Supplementary Figure 2b	21	DDX4 staining of ovaries
Supplementary Figure 4d	33	TEM
Supplementary Figure 5b	27	alpha tubulin staining of adult testis sections
Supplementary Figure 7a to f	18 & 34	negative control images for Figure 2a, 3a, 4a, 4c and 6a

Original Figure	What change(s) is(are) made
Figure 1a	unchanged, but P28 photo is now in Supplementary Figure 2c
Figure 1b	unchanged, but P28 weight data is now in Supplementary Figure 2d
Figure 1c	unchanged
Figure 1d	unchanged, now in Figure 5a
Figure 1e	PAS-H staging of seminiferous tubules has been reanalysed, now Figure 1d
Figure 1f	P28 qRT-PCR data are unchanged, now in Supplementary Figure 2e Adult qRT-PCR data are unchanged
Figure 2a	unchanged, negative control in Supplementary Figure 7a and 7b
Figure 2b	unchanged
Figure 2c	unchanged
Figure 2d	unchanged
Figure 2e	unchanged
Figure 3a	unchanged
Figure 3b	unchanged
Figure 3c	unchanged
Figure 4a	unchanged, now Supplementary Figure 3a, negative control in Supplementary Figure 7c
Figure 4b	unchanged, now Supplementary Figure 3b
Figure 4c	unchanged, now Supplementary Figure 3c
Figure 4d	unchanged, now Supplementary Figure 3d
Figure 4e	Omitted as it is a negative result
Figure 4f	unchanged, now Supplementary Figure 3e
Figure 5a	unchanged, now Figure 4a, negative control in Supplementary Figure 7d
Figure 5b	unchanged, now Figure 4b
Figure 5c	unchanged, now Figure 4c, negative control in Supplementary Figure 7e
Figure 5d	unchanged, now Figure 4d
Figure 5e	unchanged, now Supplementary Figure 4a
Figure 5f	unchanged, now Figure 4e
Figure 6a	unchanged, the graph is now Figure 5e, the images are omitted (images are originally in Supplementary Figure 2a)
Figure 6b	unchanged plus single channels have been added to the figure, now Supplementary Figure 5d
Figure 6c	unchanged plus single channels have been added to the figure, now Figure 6a, negative control in Supplementary Figure 7f
Figure 6d	unchanged plus single channels have been added to the figure, now Figure 6b, negative control in Supplementary Figure 7f
Supplementary Figure 1a to q	unchanged
Supplementary Figure 2a	unchanged, now Supplementary Figure 5a
Supplementary Figure 2b	unchanged, now Supplementary Figure 5c
Supplementary Figure 2c	unchanged, now Supplementary Figure 4c
Supplementary Figure 2d	unchanged, now Supplementary Figure 4b
Supplementary Figure 2e	unchanged, now Supplementary Figure 4c
Supplementary Figure 3a to c	unchanged, now Supplementary Figure 6

REVIEWERS' COMMENTS:

Reviewer #1 (Remarks to the Author):

I have carefully read and verified the corrections made by the authors on manuscript. The authors responded satisfactorily to this reviewer's remarks and in my opinion the paper has greatly improved and certainly more accessible to a wider audience of readers.

Reviewer #2 (Remarks to the Author):

The authors have nicely addressed most concerns from the original review. Couple comments remaining in regards to the new data and minor edits.

Major comments:

Line 280 – 282 and Sup Fig 4A - The hollow part of the HTCA seen in the KO image is not an abnormality it is the centriolar vault, which is a common presentation in TEM of the centre of the HTCA. The differences in the images presented are likely just to do with differences in sectioning. The control sperm tail is not sectioned through the middle of HTCA and tail so does not appear hollow. For detailed example images of the centriolar vault see: Fishman, E.L., Jo, K., Nguyen, Q.P.H. et al. A novel atypical sperm centriole is functional during human fertilization. *Nat Commun* 9, 2210 (2018). <https://doi-org.ezproxy.lib.monash.edu.au/10.1038/s41467-018-04678-8>

Fig 5c and e - how were the statistics conducted? It appears by the way the data is plotted that multiple individual cells from one mouse are being treated as multiple biological replicates? This is not correct. Each mouse should = 1 biological replicate regardless of how many cells were assessed for that mouse so for each genotype there should be only 4 data points plotted. Data from each mouse should be averaged and then the averages compared to assess significance, alternatively can conducted a nested t-test if you want to take into account how reliable the data is from each mouse. Any other throughout the paper like this also need to be corrected

Minor comments

Line 25. Manchette should be plural – manchettes

Line 46 result – should be results

Line 152 -153 – I would add here that it could also indicate that less round spermatids are produced (as opposed to lost). This also sets up nicely for the results of your next section.

Line 159 – The wording of this line needs tweaking. Should be 'However, approximately 30% of DDX4+ germ cells were STRA8+ in *Wdr62tm1a/tm1a* P5 testis indicating reduced or delayed meiotic initiation

Line 170 – don't need to specify in vitro and in cell culture? Should specify cell type though.

Line 218 coincide should be coincides

REVIEWERS' COMMENTS:

Reviewer #1 (Remarks to the Author):

I have carefully read and verified the corrections made by the authors on manuscript. The authors responded satisfactorily to this reviewer's remarks and in my opinion the paper has greatly improved and certainly more accessible to a wider audience of readers.

Reviewer #2 (Remarks to the Author):

The authors have nicely addressed most concerns from the original review. Couple comments remaining in regards to the new data and minor edits.

Major comments:

1. Line 280 – 282 and Sup Fig 4A - The hollow part of the HTCA seen in the KO image is not an abnormality it is the centriolar vault, which is a common presentation in TEM of the centre of the HTCA. The differences in the images presented are likely just to do with differences in sectioning. The control sperm tail is not sectioned through the middle of HTCA and tail so does not appear hollow.

For detailed example images of the centriolar vault see: Fishman, E.L., Jo, K., Nguyen, Q.P.H. et al. A novel atypical sperm centriole is functional during human fertilization. Nat Commun 9, 2210 (2018). <https://doi-org.ezproxy.lib.monash.edu.au/10.1038/s41467-018-04678-8>

Thank you for your advice and confirming that the hollow HTCA from the sperm TEM data is actually the centriolar vault. We now concluded that the HTCA is normal in our *Wdr62^{tm1a/tm1a}* sperms. Therefore, the sentences in page 11 lines 280 to 283 and Supplementary Figure 4d figure legend describing the hollow HTCA in *Wdr62^{tm1a/tm1a}* sperm TEM and how that may contribute to reduce sperm motility are now omitted. We have also corrected the labelling on Supplementary Figure 4d by adding V = centriolar vault in the figure legend and relevant image.

2. Fig 5c and e - how were the statistics conducted? It appears by the way the data is plotted that multiple individual cells from one mouse are being treated as multiple biological replicates? This is not correct. Each mouse should = 1 biological replicate regardless of how many cells were assessed for that mouse so for each genotype there should be only 4 data points plotted. Data from each mouse should be averaged and then the averages compared to assess significance, alternatively can conducted a nested t-test if you want to take into account how reliable the data is from each mouse. Any other throughout the paper like this also need to be corrected

Thank you for your comment. It was certainly not our intention to plot multiple individual cells from one mouse being treated as multiple biological replicates. The total number of cells/sperms analysed from the number of individual mice are clearly detailed in the figure legend. We do agree with reviewer 2 and have now plotted the average value from each mouse so that each plot/dot represents one biological sample in all the graphs throughout this paper. In particular, we have made this change in Figure 2a, 4d, 5c and e. As a consequence, the resulting p-values have also been updated in these figure legends.

Minor comments

3. Line 25. Manchette should be plural – manchettes

Thank you for picking up the grammatical error. We have made this change in line 25.

4. Line 46 result – should be results

Thank you for picking up the grammatical error. We have made this change in line 46.

5. Line 152 -153 – I would add here that it could also indicate that less round spermatids are produced (as opposed to lost). This also sets up nicely for the results of your next section.

Thank you for your thoughtful suggestion. We have made this suggested change in page 7 line 152 (red highlighted the change):

In contrast, there was a significantly lower expression of genes encoding Transition protein 1 (Tnp1) and Protamine 2 (Prm2), both involved in compacting the haploid genome in late round spermatids³⁷, in *Wdr62^{tm1a/tm1a}* P28 and adult testis (Figure 1f and Supplementary Figure 2e) indicating **less round spermatids are produced** and possible spermiogenesis defect in the *Wdr62^{tm1a/tm1a}* testis.

6. Line 159 – The wording of this line needs tweaking. Should be 'However, approximately 30% of DDX4+ germ cells were STRA8+ in *Wdr62^{tm1a/tm1a}* P5 testis indicating reduced or delayed meiotic initiation

Thank you for your suggestion to make this a clearer sentence. We have made this suggested change in page 8 line 159-160 (red highlighted the change):

However, approximately 30% of DDX4+ germ cells were also STRA8+ in *Wdr62^{tm1a/tm1a}* P5 testis indicating reduced or delayed meiotic initiation.

7. Line 170 – don't need to specify in vitro and in cell culture? Should specify cell type though.

Thank you for your comment. The effect of WDR62 deficiency in mitotic cell progression were studied in mouse embryonic fibroblasts from *Wdr62* mutant mice (Chen et al., 2014; Sgourdou et al., 2017), fibroblasts from human patients carrying *Wdr62* mutation (Sgourdou et al., 2017)

and during neurogenesis *in vivo* from Wdr62 loss-of-function mouse models (Chen et al., 2014; Xu et al., 2014; Jayaraman et al., 2016; Sgourdou et al., 2017). These information have been added to page 8 line 170 and 171 (red highlighted the change):

WDR62 loss-of-function has been shown to cause mitotic delay in **mouse embryonic fibroblasts, fibroblasts from patients** and during neurogenesis *in vivo*¹¹⁻¹⁴.

8. Line 218 coincide should be coincides

Thank you for picking up the grammatical error. We have made this change in line 218.